# Performance discrepancy mitigation in heart disease prediction for multisensory inter-datasets

Mahmudul Hasan[1,2,*], Md Abdus Sahid[1,*], Md Palash Uddin[1,2], Md Abu Marjan[1], Seifedine Kadry[3,4,5,6] and Jungeun Kim[7]

[1] Department of Computer Science and Engineering, Hajee Mohammad Danesh Science and Technology University, Dinajpur, Bangladesh
[2] School of Information Technology, Deakin University, Geelong, VIC, Australia
[3] Department of Electrical and Computer Engineering, Lebanese American University, Byblos, Lebanon
[4] Department of Applied Data Science, Noroff University College, Kristiansand, Norway
[5] Artificial Intelligence Research Center (AIRC), Ajman University, Ajman, Norway
[6] MEU Research Unit, Middle East University, Amman, Jordan
[7] Department of Software, Kongju National University, Cheonan, Republic of South Korea
* These authors contributed equally to this work.

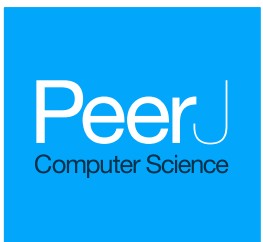

Corresponding authors
Md Palash Uddin,
palash_cse@hstu.ac.bd
Jungeun Kim, jekim@kongju.ac.kr

## ABSTRACT

Heart disease is one of the primary causes of morbidity and death worldwide. Millions of people have had heart attacks every year, and only early-stage predictions can help to reduce the number. Researchers are working on designing and developing early-stage prediction systems using different advanced technologies, and machine learning (ML) is one of them. Almost all existing ML-based works consider the same dataset (intra-dataset) for the training and validation of their method. In particular, they do not consider inter-dataset performance checks, where different datasets are used in the training and testing phases. In inter-dataset setup, existing ML models show a poor performance named the inter-dataset discrepancy problem. This work focuses on mitigating the inter-dataset discrepancy problem by considering five available heart disease datasets and their combined form. All potential training and testing mode combinations are systematically executed to assess discrepancies before and after applying the proposed methods. Imbalance data handling using SMOTE-Tomek, feature selection using random forest (RF), and feature extraction using principle component analysis (PCA) with a long preprocessing pipeline are used to mitigate the inter-dataset discrepancy problem. The preprocessing pipeline builds on missing value handling using RF regression, log transformation, outlier removal, normalization, and data balancing that convert the datasets to more ML-centric. Support vector machine, K-nearest neighbors, decision tree, RF, eXtreme Gradient Boosting, Gaussian naive Bayes, logistic regression, and multilayer perceptron are used as classifiers. Experimental results show that feature selection and classification using RF produce better results than other combination strategies in both single- and inter-dataset setups. In certain configurations of individual datasets, RF demonstrates 100% accuracy and 96% accuracy during the feature selection phase in an inter-dataset setup, exhibiting commendable precision, recall, F1 score, specificity, and AUC score. The results indicate that an effective preprocessing technique has the potential to improve the performance of the ML model without necessitating the development of intricate prediction models. Addressing inter-dataset discrepancies

introduces a novel research avenue, enabling the amalgamation of identical features from various datasets to construct a comprehensive global dataset within a specific domain.

## INTRODUCTION

Cardiovascular diseases (CVDs) are considered the most responsible diseases for death globally (*Jiang et al., 2022*; *Azmi et al., 2022*). CVDs refer to conditions affecting the heart and blood vessels. This category includes blood vessel and brain diseases, such as pulmonary embolism, coronary heart disease, deep vein thrombosis, peripheral artery disease, rheumatic heart disease, congenital heart disease, and stroke. According to the World Health Organization, in 2021 around 17.9 million people have died, and it is responsible for 32% of all fatalities (*World Health Organization, 2013*). The primary causes of the 85% total deaths are heart attacks and strokes. They cause negative effects on human health, resulting in obesity, overweight, high cholesterol, and diabetes (*Gárate-Escamila, El Hassani & Andrès, 2020*; *Rajkumar, Devi & Srinivasan, 2022*). Often, the indications of aging are puzzling, posing a challenge for practitioners when diagnosing. As heart disease is a sensitive issue, early detection could be the potential strategy to decrease heart attack rate. The two most popular diagnostic techniques for finding heart problems are electrocardiography (ECG) and coronary angiography (CA), but there is still some difference between these two. CA is costly, and ECG testing can fail to detect heart disease symptoms frequently (*Giri et al., 2013*; *Safdar et al., 2018*). In recent years, the Internet of Things (IoT) has been implemented in healthcare systems as an Internet of Medical Things (IoMT) to collect sensor data for cardiac diagnosis and prognosis (*Manimurugan et al., 2022*). Despite the delicate nature of heart disease, the accuracy of the existing dataset requires incorporating data collected by multiple sensors (*Ali et al., 2020*). Machine learning (ML) can help detect and diagnose heart disease more accurately from the available and real-time datasets (*Cutrì et al., 2017*). These techniques, along with some hidden technology, including finding relationships among the attributes within the datasets and finding more accurate techniques with high accuracy, are the primary stage of CVD classification (*Canlas, 2010*; *Helma, Gottmann & Kramer, 2000*; *Lee, Liao & Embrechts, 2000*). Recently, doctors have been using computers more to diagnose diseases. It is one of the main reasons medical data is increasing. In this way, ML is becoming an essential tool for diagnosing health problems in the 21$^{st}$ century. It is used in cases where *corpus* data is difficult to program and manually review, such as genetic data analysis, pandemic prediction, and the transformation of medical data into knowledge (*Sarumi & Leung, 2022*; *Tiwari et al., 2022*; *Weissler et al., 2021*). Researchers from various fields are doing a wide range of research to predict cardiac diseases using datasets collected from the UCI ML repository. Most studies are conducted on a single dataset, and some tried to

combine two or three related datasets for analysis (*Subramaniyam, Mahapatra & Singh, 2019*; *Shah et al., 2017*; *Spencer et al., 2020*; *Gárate-Escamila, El Hassani & Andrès, 2020*). None of them consider the inter-dataset discrepancy problem (*Lin et al., 2018*) while working with more than one dataset. To develop a real-time prediction system, we need to develop an ML model trained by different data, making it more versatile and more general for new input sensor data directly from the human body. To create a global model from the existing heart disease prediction dataset, we need to mitigate the raised inter-dataset discrepancy problem. After mitigating this issue, we can train the ML models by the datasets, and this model is flexible to real-time multiple sensor data during the prediction time. When we consider multiple datasets in a single analysis, we train the model using different datasets and test it using different combinations. During this kind of analysis, a problem degrades the classifiers' performance (*Brynjarsdóttir & O'Hagan, 2014*). It is called an "inter-dataset discrepancy problem", where one or more datasets are used for training, and another dataset is used for testing purposes. It is also challenging to relate the datasets together to form a general dataset in this field like the sensor observed data from the human body (*Dahiya et al., 2021*). Selecting features is a key challenge for problems containing many attributes (*Rajkumar & Reena, 2010*). The proposed approach combines imbalance data handling, dimensionality reduction (feature extraction and feature selection), and the most popular ML classifiers to classify CVDs in the inter-dataset setup. In such a case, the SMOTETomek imbalance data handling technique (*Ijaz, Attique & Son, 2020*) shows better results and reduces the computational cost. RF is the best suggestion for selecting the features. In addition, different training and testing modes are performed, where one or more than one dataset is used as the training part, and another one or more than one dataset is used as the testing part. On top of that, all possible combinations are tested before prepossessing and after prepossessing. We find a significant variation in the results before and after prepossessing the data. The proposed preprocessing techniques with the classifiers mitigate the inter-dataset discrepancy problem, and among the classifiers, RF performs better than others. Further, divergent performance assessment strategies are employed to properly assess the classifiers' effectiveness. The primary objective of this study is to address inter-dataset discrepancy challenges without resorting to the creation of intricate classification models. Our focus is on identifying a more adept preprocessing pipeline capable of enhancing the performance of the ML model, all while maintaining the existing model architecture. This impetus drives us to devise a resilient preprocessing pipeline instead of delving into the complexity of model development. To this end, a summary of this article's significant contributions is listed below:

- We analyze the performance discrepancy considering the inter-dataset setup in heart disease prediction.
- We propose the potential preprocessing techniques, including RF (a more ML-centric method), PCA (the most common and popular feature extraction method), and SMOTETomek as an imbalanced data handling technique to mitigate the inter-dataset discrepancy.

- We investigate the simple but effective ML classification algorithms in our proposed inter-dataset setup and analyze their performance to predict heart disease.

The following parts of this study are structured: In "Literature Review", we look at the previous works in this field and create a comparison chart of the literature. Dataset preprocessing and a description of the datasets are provided in "Dataset Description, Recreation, and Combined Dataset Formation". In "Method", we examine the methods and materials. "Result and Analysis" contains the analysis and discussion of the results. Last, "Conclusion and Future Work" provides the conclusion and future work.

## LITERATURE REVIEW

Many researchers are still working on different methods for predicting heart disease on different datasets. Most used a single dataset in their analysis, and some used two or three datasets independently. Statistical tools, ML, deep learning, and ensemble and hybrid techniques are used to improve classification accuracy. Also, feature extraction and selection techniques are still used to improve prediction accuracy. Throughout this portion, we discuss recent studies in this field for various datasets.

### Previous works on Cleveland dataset

*El-Hasnony et al. (2022)* applied Active Learning-based ML to predict heart disease. This study employed MMC, Random, Adaptive, QUIRE, and AUDI as selection strategies, which also reduce labeling costs for multi-label active learning. They evaluated their proposed experiment regarding performance metric accuracy and the f1-score with or without hyperparameter tuning and cross-validation. Regarding accuracy, the adaptive method advanced others, scoring $0.514 \pm 0.032$ before and $0.574 \pm 0.020$ after hyperparameter optimization. Also, for the f1-score, the adaptive method showed its superiority for the heart disease prediction score of $0.623 \pm 0.040$ before and $0.6062 \pm 0.036$ after hyperparameter tuning. *Selvi & Muthulakshmi (2021)* in 2021 developed an optimal artificial neural network (OANN) by combining distance based misclassified instance removal (DBMRI), teaching and learning based optimization (TLBO), and artificial neural network (ANN). DBMRI is used to extract incorrectly classified instances from the data, and TLBO is an optimization algorithm. Moreover, the model is evaluated utilizing the Cleveland dataset in the offline phase. Real-time data is cleaned, preprocessed, and streamed to the model using Apache Spark in the online phase to detect the potentiality of heart problems. During the test phase, the OANN model surpassed other models by scoring 95.31% accuracy. *Motarwar et al. (2020)* also proposed an ML-based cognitive technique for predicting heart disease based on the Cleveland heart disease dataset. The stated study made use of five classifiers: logistic model tree (LMT), SVM, Hoeffdpin decision tree (HDT), RF, as well as GNB. They applied feature selection approaches to their work. GNB had a 93% accuracy prediction, SVM had a 90% prediction, RF had a 95% prediction, HDT had an 81% prediction, and LMT had an 81% prediction. *Mohan, Thirumalai & Srivastava (2019)* used the Hybrid Random Forest Linear Model (HRFLM) on the Cleveland dataset to obtain 88.7% accuracy. They also tried out KNN,

DT, genetic algorithm (GA), and naive Bayes (NB), among other machine-learning models. HRFLM performed the best out of all of them. *Hasan et al. (2018)* proposed a performance analysis of classification approaches to predict heart disease. They employed the Cleveland heart disease dataset. Five different classifiers, namely KNN, DT (ID3), GNB, LR, and RF are used. Using 14 attributes, KNN provided 70%, DT (ID3) provided 88%, GNB provided 88%, LR provided 89%, and RF provided 89% accuracy. After performing RF feature selection, the ten most important features are selected. KNN provided 71%, DT (ID3) provided 91%, GNB provided 91%, LR provided 93%, and RF provided 92% accuracy. In 2017, *Uyar & İlhan (2017)* presented a method for predicting heart disease utilizing a GA-based Recurrent fuzzy neural network (RFNN). Their proposed model, GA-based RFNN, utilizes the Cleveland heart disease dataset. Their proposed model predicted 98% accuracy. *Dun, Wang & Majumder (2016)* in 2016 used RF, LR, SVM, and neural networks (NN) to evaluate the prevalence of heart disease. They also performed hyperparameter tuning and feature selection. RF provided 77%, LR provided 75%, linear SVM provided 75%, SVM provided 69%, and NN provided 78% accuracy. Their study also utilized the benchmark Cleveland heart disease dataset.

## Previous works on UCI heart disease dataset

For heart disease prediction, *Bharti et al. (2021)* coupled ML and deep learning algorithms. They achieved an accuracy of 83% using LR, 85% using KNN, 83% using SVM, 80% using RF, 82% using DT, and 94% using deep learning by employing the UCI heart disease dataset and also Lasso as a feature selection technique. In 2021, *Katarya & Meena (2021)* compared nine different ML and deep learning classifiers for heart disease prediction. Two different datasets from the UCI repository were utilized for their study. Also, data were preprocessed by doing normalization, missing value replacement, and feature selection. Moreover, state-of-the-art RF performed the best, achieving 95.60% accuracy, followed by LR, SVM, and artificial neural networks (ANN). *Mehmood et al. (2021)* developed Cardio Help. This framework uses a convolutional neural network (CNN) to determine the chance of a subject developing CVD by utilizing the heart disease dataset from the UCI repository. Furthermore, the LASSO technique was applied to pick and normalize relevant features. On top of that, two distinct classifiers were created, one for binary and the other for four classes. The binary classifier achieved 97% accuracy, whereas the four-class classifier achieved 86% accuracy. In 2020, *Farzana & Veeraiah (2020)* used several different ML and data mining classification techniques, such as GNB, SVM, RF, KNN, and XGBoost. They employed the UCI heart disease dataset, SVM provided 82%, RF provided 89%, GNB provided 82%, KNN provided 67%, and XGBoost provided 79% accuracy. In 2019, *Alotaibi (2019)* presented the utilization of an ML model to identify heart disease based on the UCI heart disease dataset and applied DT, LR, RF, NB, and SVM as classifiers. The DT provided 93%, LR provided 87%, RF provided 89%, NB provided 87%, and SVM provided 92% prediction accuracy. *Repaka, Ravikanti & Franklin (2019)* developed sophisticated heart disease estimations using the UCI dataset. They implemented four different techniques; among them, NB gave the best accuracy of 89.77%.

## Previous works on other single datasets

*Dritsas & Trigka (2023)*, proposed a ML-based CVD risk prediction model. They balanced the imbalanced CVD dataset using the Synthetic Minority Oversampling Technique (SMOTE). In the context of feature selection, they employed three feature selection methods: RF, information gain, and gain ratio. Also, they employed several classifiers for CVD prediction and evaluated their experimental results with or without the application of SMOTE with 10-fold cross-validation and compared them using different performance metrics. The study outcomes revealed that the Stacking ensemble model followed SMOTE with 10-fold cross-validation, advancing of others, reaching an accuracy of 87.8%, a recall of 88.3%, a precision of 88%, and an AUC of 98.2%. *Manimurugan et al. (2022)* presented a two-stage heart disease prediction model. Its first step categorized data collected from medical sensors attached to the subject's body; the second stage classified echocardiography images to predict heart disease. Furthermore, a hybrid linear discriminant analysis with enhanced ant lion optimization (HLDA-MALO) approach was utilized for sensor data identification. Moreover, a hybrid Faster R-CNN with SE-ResNet-101 model was applied for echocardiography image classification. The HLDA-MALO approach detected healthy sensor data with 96.85% accuracy and abnormal sensor data with 98.31% accuracy. Their proposed hybrid Faster R-CNN with SE-ResNeXt-101 transfer learning model performed better in identifying echocardiography pictures, with 98.06% precision, 98.95% recall, 96.32% specificity, a 99.02% F-score, and 99.15% maximum accuracy. *Chowdhury et al. (2019)* proposed heart disease prediction using a smart digital stethoscope system. They used phonocardiogram (PCG) signal data containing normal and abnormal heart sounds for this analysis. They preprocessed the signal data, segmented it, and extracted different features from the segmented data, which will be fed to ML classifiers. The DT, discriminant analysis, SVM, KNN, and ensemble classifiers with five-fold cross-validation are employed for heart disease prediction. The cost-adjusted optimal ensemble approach can diagnose abnormal and normal HS with 97% and 88% accuracy, respectively, among employed classifiers. *Raza (2019)* attempted to forecast heart disease in 2019 using the ensemble learning method utilizing the Statlog heart disease dataset available in the UCI ML repository. They used a majority voting rule to combine LR, NB, and ANN. This approach had an accuracy of 88.88%. In 2019, *Beunza et al. (2019)* proposed comparing ML classifiers for diagnostic coronary heart disease prediction. This study used a dataset from the Framingham heart study. Also, they employed six classifiers: DT, Boosted DT, SVM, NN, RF, and LR. DT provided 84%, Boosted DT provided 84%, RF provided 74%, SVM provided 68%, NN provided 71% and LR provided 66% accuracy according to R-Studio Model. *Chen et al. (2007)* used SVM, NN, BM, DT, and LR to predict heart disease. They used a dataset collected from AnZhen Hospital in Beijing, China. The state-of-the-art SVM provided 91%, NN provided 89%, BM provided 82%, DT provided 78% and LR provided 74% accuracy.

## Previous works on multiple datasets

*Valarmathi & Sheela (2021)* proposed hyperparameter optimization to predict heart disease in 2021. They used Grid Search (GS), Randomized Search (RS), and TPOT

classifiers for optimizing the performance of RF and XGBoost classifiers. For this analysis, they used the Cleveland and Z-Alizadeh Sani datasets. Upon the Cleveland heart disease dataset, RF with GS provided 91% accuracy, RS provided 95% accuracy, and TPOT provided 98.5% accuracy. On the other hand, the XGBoost algorithm with GS provided 86%, RS provided 92% and TPOT provided 91% accuracy on the Cleveland heart disease dataset. However, on the Z-Alizadeh Sani dataset, RF with RS provides the highest 80%, 74%, and 77% accuracy for the three vessels. *Rathi et al. (2021)* in 2021 used the C4.5 algorithm to build a prediction model on the UCI heart problems dataset from Kaggle to predict the likelihood of heart disease. Their proposed model achieved 89% accuracy. *Mienye, Sun & Wang (2020)* used an ensemble technique for cardiac disease prediction. Both the Framingham and Cleveland datasets were used. Using CART, they created a homogenous ensemble AB-WAE. In addition, they employed several classifiers. Their employed LR presented 78%, LDA presented 78%, KNN presented 60%, SVM presented 79%, GB presented 81%, RF presented 83%, CART presented 68%, and the presented AB-WAE presented 93% accuracy using the Cleveland heart disease dataset. *Ayon, Islam & Hossain (2020)* investigated numerous ML approaches for heart disease prediction, utilizing the StatLog and Cleveland datasets separately. Also, two distinct *K*-values were used to split the data (5,10). On the StatLog dataset, SVM had a five-fold classification accuracy of 97% and a ten-fold classification accuracy of 95%. On top of that, SVM scored 97% for five-fold classification and 95% for ten-fold classification on the Cleveland dataset.

*Khan (2020)* built an IoT system to alert physicians to the possibility of a patient developing heart disease. The system is powered by an ML approach trained using the UCI, Framingham, and Public Health datasets. They chose the relevant features using the MCFA method. After that, the key features are used to train MDCNN and other models after they have been selected. The models, once trained, use sensor data to predict the likelihood that the subject will develop heart disease. As a result, MDCNN outperforms all other models in terms of overall performance. *Gárate-Escamila, El Hassani & Andrès (2020)* in 2020 applied different ML algorithms along PCA and Chi-square on the UCI heart disease dataset. They used Cleveland, Hungarian, and Cleveland-Hungarian combined datasets. As a consequence, they got a maximum accuracy of 99% for RF, LR, and Gradient boosting classifier (GBC), 98% for DT, 87% for MLP, and 69% for NB. In 2020, *Spencer et al. (2020)* proposed a feature selection and classification method for heart disease prediction. They created a mixed dataset for their analysis, combining data from Cleveland, Long Beach, VA, Hungary, and Switzerland. This combined dataset contains 14 features with 720 samples. Moreover, they applied Bayes Net (BN), LR, Stochastic Gradient Descent (SGD), Instance-Based Learner (IBK), AdaBoost, JRip, and RF. The BN algorithm provided a maximum of 85% accuracy on the original combined dataset. After that, they also applied PCA, Chi-square, ReliefF, and Symmetrical Uncertainty (SU) on the combined dataset. IBK algorithm provided a maximum 84% accuracy on the Heart-PCA dataset with 11 features. The BN algorithm provided a maximum of 85% accuracy on the Heart-Chi-Square dataset with ten features. On the other hand, the SGD algorithm provided a maximum of 85% accuracy on the Heart-RelifF dataset, and BN provided a maximum of 85% accuracy on the Heart-SU dataset. *Reddy et al. (2019)* employed KNN,

**Table 1 Comparison of the related literature on the different datasets.**

| Dataset | Ref. | Models | Performance | Year |
|---|---|---|---|---|
| Cleveland | El-Hasnony et al. (2022) | MMC, Random, Adaptive, QUIRE, and AUDI as selection strategies of Active Learning | In terms of accuracy, scoring $0.514 \pm 0.032$ before and $0.574 \pm 0.020$; Also, for the f1-score, the adaptive method showed a prediction score of $0.623 \pm 0.040$ before and $0.6062 \pm 0.036$ after hyperparameter tuning. | 2022 |
| Cleveland | Selvi & Muthulakshmi (2021) | ANN with TLBO and DBMRI | ANN with TLBO and DBMRI (95.41), TLBO-ANN (90.75), J48 (78.55), Random Tree (72.94), RBF Network (83.49), NBTree (79.21), RF (82.18) | 2021 |
| Cleveland | Motarwar et al. (2020) | RF, GNB, SVM, Hoeffding DT and Logistic Model Tree (LMT) | HRFLM (88.7), DT (85), NB (75.8) | 2020 |
| Cleveland | Mohan, Thirumalai & Srivastava (2019) | HRFLM, KNN, DT, GA, NB | GNB 93%, SVM 90%, RF 95%, Hoeffding DT 81% and LMT 81% | 2019 |
| Cleveland | Hasan et al. (2018) | KNN, DT (ID3), GNB, LR and RF | Using 14 attributes KNN 70%, DT (ID3) 88%, GNB 88%, LR 89% and RF 89%. Using 10 attributes KNN 71%, DT (ID3) 91%, GNB 91%, LR 93% and RF 92%. | 2018 |
| Cleveland | Uyar & İlhan (2017) | GA based RFNN | RFNN 98% | 2017 |
| Cleveland | Dun, Wang & Majumder (2016) | RF, LR, Linear SVM, RBF SVM and NN | LR 83%, KNN 85%, SVM 83%, RF 80%, DT 82% and Deep Learning 94% | 2016 |
| UCI heart disease dataset | Bharti et al. (2021) | LR, KNN, SVM, RF, DT and Deep Learning | LR 83%, KNN 85%, SVM 83%, RF 80%, DT 82% and Deep Learning 94%. | 2021 |
| UCI heart disease dataset | Katarya & Meena (2021) | RF, LR, SVM, ANN, NB, KNN, DT, DNN, MLP | RF (95.60), LR (93.40), SVM (92.30), ANN (92.30), NB (90.10), KNN (71.42), DT (81.31), DNN (79.92), MLP (75.42) | 2021 |
| UCI heart disease dataset | Mehmood et al. (2021) | CNN | 97% for binary classification, 86% for quaternion classification | 2021 |
| UCI heart disease dataset | Farzana & Veeraiah (2020) | GNB, SVM, RF, KNN and XGBoost | GNB 82%, SVM 82%, RF 89%, KNN 67% and XGBoost 79% | 2020 |
| UCI heart disease dataset | Alotaibi (2019) | DT, LR, RF, NB and SVM | DT 93%, LR 87%, RF 89%, NB 87% and SVM 92% | 2019 |
| UCI heart disease dataset | Repaka, Ravikanti & Franklin (2019) | NB, SMO | NB 89.77%, SMO 84.77%, Bayes Net 81.11%, MLP 77.4% | 2019 |
| StatLog | Raza (2019) | Ensemble (LR, NB, ANN) | Ensemble Algorithm 88.88% | 2019 |
| Framingham | Beunza et al. (2019) | DT, Boosted DT, RF, SVM, NN and LR | SVM 91%, NN 89%, BM 82%, DT 78% and LR 74%. | 2007 |
| CVD dataset | Dritsas & Trigka (2023) | NB, LR, MLP, NN, RF, Rotation Forest, AdaBoost, Stacking, Bagging, and Voting | Among these algorithms stacking ensemble advancing of others. The accuracy of 87.8%, a recall of 88.3%, a precision of 88%, and an AUC of 98.2% provided a stacking ensemble. | 2023 |
| Sensor and Image dataset | Manimurugan et al. (2022) | HLDA-MALO, and hybrid Faster R-CNN with SE-ResNet-101 model | The HLDA-MALO approach detected healthy sensor data with 96.85% accuracy and abnormal sensor data with 98.31% accuracy. The hybrid Faster R-CNN with SE-ResNeXt-101 transfer learning model identifying echocardiography pictures, with 98.06% precision, 98.95% recall, 96.32% specificity, a 99.02% F-score, and 99.15% maximum accuracy. | 2022 |

**Table 2 Comparison of the related literature based on different datasets.**

| Dataset | Ref. | Models | Performance | Year |
|---|---|---|---|---|
| PCG Signal dataset | *Chowdhury et al. (2019)* | DT, discriminant analysis, SVM, KNN, and ensemble classifiers | Ensemble approach is capable of diagnosing abnormal and normal HS with 97% and 88% accuracy, respectively | 2019 |
| Collected from AnZhen Hospital, Beijing | *Chen et al. (2007)* | SVM, NN, BM, DT and LR | DT 84%, Boosted DT 84%, RF 74%, SVM 68%, NN 71% and LR 66% according to R-Studio model C. | 2019 |
| Cleveland and Z-Alizaeh Sani dataset | *Valarmathi & Sheela (2021)* | RF, XGBoost | RF 91%, XGBoost 92% | 2021 |
| Cleveland, Hungary, Switzerland, and the VA Long Beach | *Rathi et al. (2021)* | C4.5 | C4.5 89% | 2021 |
| Framingham and Cleveland dataset | *Mienye, Sun & Wang (2020)* | KNN, LR, LDA, SVM, CART, GB, RF and ensemble AB-WAE | KNN 60%, LR 78%, LDA 78%, SVM 79%, CART 68%, GB 81%, RF 83% and AB-WAE 93%. | 2020 |
| Statlog, Cleveland | *Ayon, Islam & Hossain (2020)* | SVM, LR, DNN, DT, NB, RF, KNN | Best accuracy SVM (97% fivefold) | 2020 |
| UCI, Farmingham, Public Health, Sensor Data | *Khan (2020)* | MDCNN, LR, DLNN | MDCNN (93.3%), LR (87.8%), DLNN (81.8%) MDCNN (98.2%), LR (88.8%), DLNN (83.8%) MDCNN (97.6%), LR (84.6%), DLNN (81.6%) MDCNN (96.30%), LR (83.6%), DLNN (82.4%) | 2020 |
| Cleveland, Hungarian | *Gárate-Escamila, El Hassani & Andrès (2020)* | RF, LR, GBC, DT, MLP, and NB | 99% for RF, LR and GBC, 98% for DT, 87% for MLP and 69% for NB | 2020 |
| Cleveland, Long-Beach-VA, Hungarian and Switzerland dataset | *Spencer et al. (2020)* | Bayes Net, LR, SGD, IBK, AdaBoost, JRip, and RF | Bayes Net 85% (Maximum) | 2020 |
| Cleveland And Statlog | *Reddy et al. (2019)* | KNN, SVM, RF, NB and NN | Using 6 features KNN 86%, SVM 83%, RF 91%, NB 87% and NN 86% | 2019 |
| BIDMC + MIT BIH Arrhythmia databases | *Masetic & Subasi (2016)* | RF, ANN, SVM, KNN, C4.5 | RF 100%, ANN 99.32%, SVM 99.96%, KNN 99.75%, C4.599.86% | 2016 |
| Undefined | *Krishnan & Geetha (2019)* | DT and NB | RF 100%, ANN 99.32%, SVM 99.96%, KNN 99.75%, C4.599.86% | 2019 |
| Wearable and EMR data | *Ali et al. (2021)* | SVM, LR, MLP, RF, DT, NB, Ensemble | SVM 84%, LR 92%, MLP 85%, RF 87%, DT 78%, NB 83% and their proposed ensemble 99% | 2021 |

SVM, RF, NB, and NN classifiers on the Cleveland and StatLog heart disease datasets for prediction. They chose six and eight features using methods for selecting features. Using six features, KNN 86%, SVM 83%, RF 91%, NB 87%, and NN 86% accuracy was displayed. Models with six features have a marginally higher average accuracy than those with eight. *Masetic & Subasi (2016)* in 2016 carried out the research by using Electrocardiogram reports to make a heart disease prediction model. The Autoregressive Burg method was used to extract features. A total of three datasets were used, collected from three different sources, two datasets contained data from people with congestive heart failure, and one dataset had data from people with normal conditions. Several classifiers were used to get the best-performing model. Among those used classifiers, RF had the best accuracy of

**Table 3  Population of patients and controls in each dataset (in number and %).**

| Dataset | No. of observations | No. of patients | No. of controls | % of Patients | % of Controls |
|---|---|---|---|---|---|
| Cleveland | 303 | 165 | 138 | 54.46 | 45.54 |
| Hungarian | 294 | 106 | 188 | 36.05 | 63.95 |
| Switzerland | 123 | 115 | 8 | 93.50 | 6.50 |
| Long Beach | 200 | 149 | 51 | 74.50 | 25.50 |
| Stat Log | 270 | 120 | 150 | 44.44 | 55.56 |
| Combined | 918 | 508 | 410 | 55.34 | 44.66 |

100% in both dataset combinations. A group of researchers employed two different ML techniques, DT and NB. The DT presented 91%, and the NB presented 87% accuracy. In 2021, *Ali et al. (2021)* developed a smart heart disease prediction approach utilizing feature fusion and ensemble deep learning. Both the sensor information and data from electronic medical records are used. They extracted significant features from those datasets and merged them. Moreover, they used LogitBoost as a base learner for the proposed ensemble mode. Their fusion dataset with specific feature weighting method provides accuracy 84% for SVM, 92% for LR, 85% for MLP, 87% for RF, 78% for DT, 83% for NB, and 99% for their proposed ensemble. A comparison of related literature is tabulated in Tables 1 and 2.

# DATASET DESCRIPTION, RECREATION, AND COMBINED DATASET FORMATION

## Dataset description

In this study, the well-known five heart disease datasets, namely Cleveland (*Janosi et al., 1988*), Hungarian (*Janosi et al., 1988*), Switzerland (*Janosi et al., 1988*), Long Beach VA (*Janosi et al., 1988*), StatLog (*Statlog (Heart), 2024*), and a combined dataset formed using these five datasets have been experimented with. The selection of these five datasets for the experiment is based on the similarity of features. Those five datasets are publicly available, and most of the research works in this domain are based upon the datasets, as the literature review sections justify. total observations include patients and controls, with percentages, as provided in Table 3. Notice that in our combined dataset, there are 918 observations after removing the 272 duplicates. We are interested in how often each patient and control group appear. In such a scenario, it is evident that the datasets are imbalanced, which is a crucial factor in diminishing the accuracy of classifiers. The study also analyzes how balancing the datasets might improve performance. Moreover, all the datasets, including the combined one, contain 11 common features (Age, Sex, ChestPainType, RestingBP, Cholesterol, FastingBS, RestingECG, MaxHR, ExerciseAngina, Oldpeak, and ST_Slope) and a target feature (heart disease). On top of that, the features are described in Table 4 with their data types.

## Dataset recreation

It is a crucial stage to reform the datasets based on a threshold for working with different datasets (*Stefenon et al., 2022*). In this study, the other four datasets are recreated based on

**Table 4 Common features of the datasets.**

| Feature name | Feature type | Detail |
|---|---|---|
| Age | Integer | Age of the patient. Which is expressed in the year. |
| Sex | String | M-Male and F-Female. |
| ChestPainType | String | TA-Typical Angina, ATA-Atypical Angina, NAP-Non-Anginal Pain, ASY-Asymptomatic. |
| RestingBP | Integer | Resting Blood Pressure in mmHg. |
| Cholesterol | Integer | Serum Cholesterol in mm/dl. |
| FastingBS | Binary | $1$ -if Fasting BS $> 120$ $mg/dl$, $0 - otherwise$. |
| RestingECG | String | Resting Electrocardiogram results (Normal: Normal, ST: having ST-T wave abnormality (T wave inversions and/or ST elevation or depression of $> 0.05$ mV), LVH: showing probable or definite left ventricular hypertrophy by Estes' criteria. |
| MaxHR | Integer | Maximum Heart Rate achieved in between 60 and 202. |
| ExerciseAngina | Char/ String | Exercise-Induced Angina. Y-Yes, N-No. |
| Oldpeak | Float | Stress Test-ST depression induced by exercise relative to rest. |
| ST_Slope | String | The slope of the peak exercise ST segment. Up-upsloping, Flat-flat, Down-downsloping |
| Heart disease | Binary | Output class: 1-heart disease and 0-Normal. |

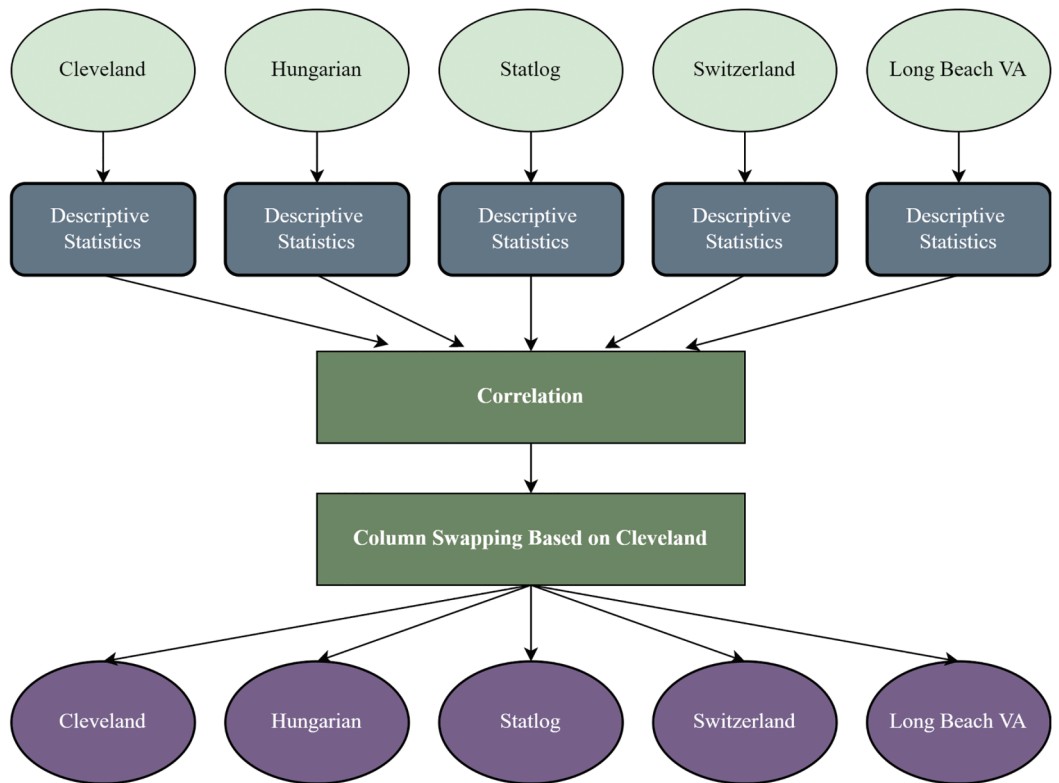

**Figure 1 Dataset recreation through column swapping by the internal characteristics of the dataset.**

**Table 5 Combinations on training and testing datasets to check the inter-data discrepancy.**

| Comb. | Training set | Testing set | Comb. | Training set | Testing set |
|---|---|---|---|---|---|
| 1 | Cleveland | Statlog | 16 | Statlog, Long Beach VA, Hungarian, Switzerland | Cleveland |
| 2 | Statlog | Long Beach VA | 17 | Cleveland, Long Beach VA, Hungarian, Switzerland | Statlog |
| 3 | Long Beach VA | Hungarian | 18 | Cleveland, Long Beach VA, Switzerland, Statlog | Hungarian |
| 4 | Hungarian | Switzerland | 19 | Cleveland, Statlog, Hungarian, Switzerland | Long Beach VA |
| 5 | Long Beach VA | Cleveland | 20 | Cleveland, Statlog, Long Beach VA, Hungarian | Switzerland |
| 6 | Cleveland, Statlog | Long Beach | 21 | Cleveland, Statlog | Long Beach VA, Hungarian |
| 7 | Cleveland, Statlog | Hungarian | 22 | Statlog, Long Beach VA | Cleveland, Hungarian |
| 8 | Hungarian, Long Beach VA | Statlog | 23 | Long Beach VA, Switzerland | Statlog, Hungarian |
| 9 | Long Beach VA, Hungarian | Switzerland | 24 | Cleveland, Statlog | Hungarian, Switzerland |
| 10 | Statlog, Long Beach VA | Cleveland | 25 | Statlog, Hungarian | Cleveland, Long Beach VA |
| 11 | Cleveland, Statlog, Long Beach VA | Hungarian | 26 | Cleveland, Statlog, Long Beach VA | Hungarian, Switzerland |
| 12 | Statlog, Long Beach VA, Hungarian | Cleveland | 27 | Statlog, Long Beach VA, Hungarian | Cleveland, Switzerland |
| 13 | Statlog, Long Beach VA, Hungarian | Switzerland | 28 | Statlog, Cleveland, Switzerland | Long Beach VA, Hungarian |
| 14 | Long Beach VA, Hungarian, Switzerland | Statlog | 29 | Cleveland, Hungarian, Switzerland | Statlog, Long Beach VA |
| 15 | Cleveland, Hungarian, Statlog | Long Beach VA | 30 | Long Beach VA, Hungarian, Switzerland | Cleveland, Statlog |

the Cleveland dataset's sequence of attributes. After recreating the datasets, every dataset's column is identical and contains the same attributes and data types. Figure 1 shows the overview of the recreation of the datasets. We find each dataset's descriptive statistics (mean median, standard deviation, kurtosis, and skewness).

## Combined dataset formation

The recreated datasets are merged one after another to form the combined one. However, the combined dataset contains some duplicate values, and after removing the duplicates, the final combined dataset is created.

## Combination to check the inter-dataset performance

To check the inter-dataset discrepancy in heart disease prediction, we created 30 combinations using the five individual datasets with 1:1, 2:1, 3:1, 4:1, 2:2, and 3:2 ratios. In addition, Table 5 tabulates all the combinations of the datasets for analyzing the inter-dataset performance.

# METHOD

## Method overview

The methodology is divided into four major segments: (i) data preprocessing, (ii) dimensionality reduction, (iii) model building using ML classifiers, and (iv) model

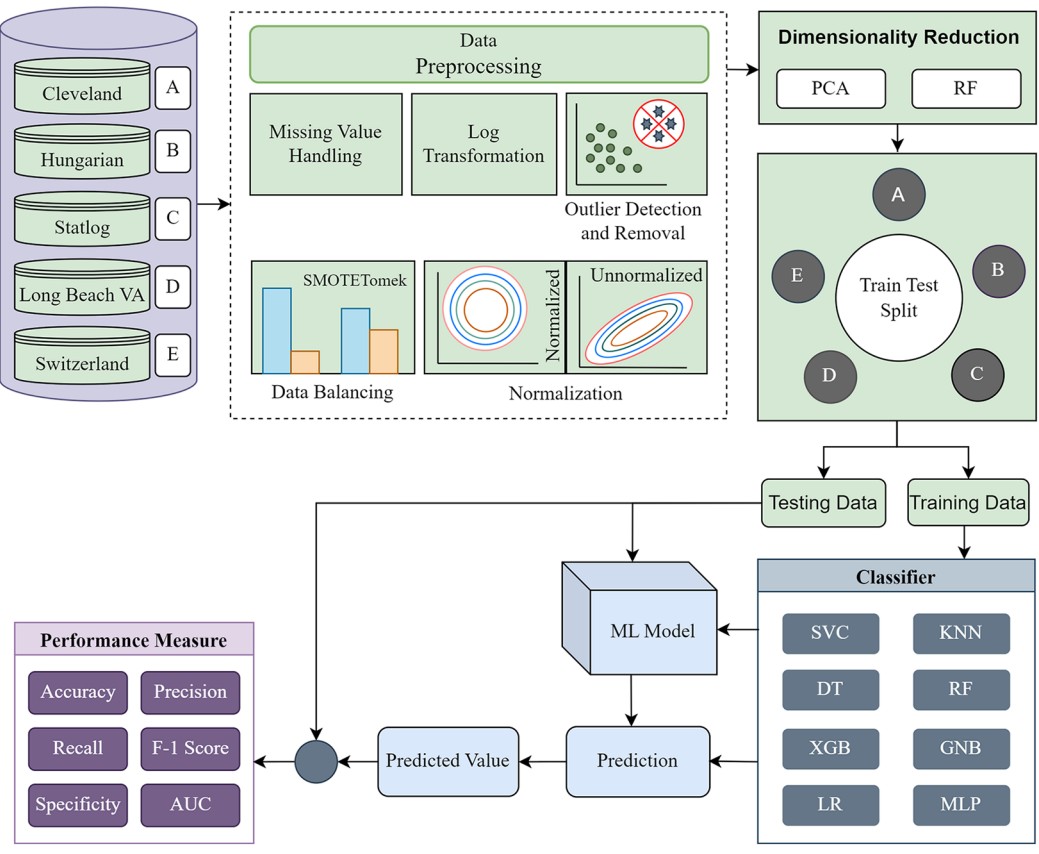

**Figure 2 Proposed methodology to mitigate the inter-dataset performance discrepancy in heart disease prediction.** Basic data preprocessing with dimensionality reduction using PCA and RF for the individual datasets and combining the datasets in various permutations.

evaluation. First, each of the five individual datasets is investigated to determine their statistical characteristics. Based on the characteristics, different preprocessing techniques are applied to convert the datasets into more ML-centric to fit the ML classifiers accurately. We recreate the datasets into the same form based on the columns of the Cleveland dataset. Following the basic preprocessing, we apply PCA and RF separately for single and combined datasets and various dataset combinations. The performance of each classifier for different phases is tabulated and visualized by different plots to find the inter-dataset discrepancy and how the proposed methodology solves the problem. The proposed methodology's block diagram is depicted in Fig. 2.

## Data preprocessing

For the analysis of the datasets, key preprocessing is mandatory. The total preprocessing techniques are divided into two different phases. The first phase contains missing value handling, log transformation, outlier detection, normalization, and imbalanced data handling. In the second phase, we reduce the features using two popular techniques, PCA as feature extraction and RF as feature selection.

---

**Algorithm 1** Missing value handling.

**Input:** $\theta$ (Regression model); $\xi$ (Independent variables); $v$ (Missing value columns)

**Output:** Suitable values for the missing values

1:    **procedure** *MissingValueHandle ($\theta$, $\xi$, $v$)*

2:       **for** each $v$ **do**

3:          $P^v \leftarrow Predict(\theta, \xi)$;

            [$P^v$ is the predicted values for $v$]

4:       **end for**

5:    **end Procedure**

---

**Missing value handling:** There is often a missing value in real-world datasets. It is the cause of the skewed outcome and has a major effect on how well ML algorithms perform (*Thomas et al., 2020*). Addressing the problem of missing data values in datasets is crucial for increasing prediction precision. Dropping missing data or filling it in with averages, medians, and point estimates are two methods for dealing with missing data (*Zhang, 2016*). The datasets used in this study contain many missing values that must be filled out. Because of the limited sample sizes of the datasets, it is infeasible to delete rows to process the values. In addition, most values have a property that allows us to replace the missing values gap by averaging the columns, which yields better performance than alternative handling methods. However, there are many missing values in some columns in Switzerland, Hungarian, and Long Beach VA datasets, and the traditional approach is inappropriate to handle these cases. So, we consider the missing value column as the dependent variable and other correlated columns as independent variables to solve this issue. As a result, we employ the RF regression model to impute missing values with suitable replacements, leveraging its robustness against overfitting and less sensitive nature to outliers. The pseudocode in Algorithm 1 illustrates the missing value-handling process. After that, the distribution of the data exhibits some properties of a normal distribution, and to transform it into a more normal distribution, we employ a log transformation.

**Outlier detection:** When building and deploying ML algorithms, outlier detection is crucial. However, outliers in a dataset reduce the performance of the algorithms (*Ramaswamy, Rastogi & Shim, 2000*), and it is detected using Turkey fences (*Zhou et al., 2006*). Since Turkey fences are suitable for datasets with a normal distribution, we employ them in our preprocessing pipeline to handle outliers. The dataset is split into three quartiles: Q1, Q2, and Q3. The first quartile, or Q1, is the value within the data set comprising 25% values below it. The third quartile, or Q3, is indeed the value that accounts for 25% of the values above it. Outliers are also valued below or above the lower or upper limits, as calculated below.

$lower\_limit = Q_1 - 1.5(Q_3 - Q_1)$
$upper\_limit = Q_1 + 1.5(Q_3 - Q_1)$

Outliers that fall below the lower limit are replaced with a lower limit, and outliers above the upper limit are replaced with an upper limit.

**Normalization:** Normalization converts the numerical column values of a dataset to a standard scale (*García, Luengo & Herrera, 2015*). It is important when an ML model employs Euclidean distance to interpret the inputs (*Taunk et al., 2019*). Since we transform our data to be normally distributed and remove outliers, both Min-max normalization and Standardization can work well. So, this work transforms the datasets into normalized ones using the Min-Max scaling method. Therefore, we employ the Min-max scaling method. It subtracts the smallest value from the maximum value of the column and divides it by the range. After normalization, each column's value falls between 0 and 1.

**Imbalance data handling:** One drawback of imbalanced data is that insufficient data from the underrepresented group prevents the algorithm from learning an appropriate cutoff point (*Norori et al., 2021*). Under-sampling and over-sampling are the two most common approaches to solving this issue. Replicating random samples from the minority group is an example of oversampling while selecting random instances from the majority group and removing them is an example of undersampling. This investigation made use of SMOTETomek ensemble methods to cope with data imbalance (*Hasan et al., 2022*). SMOTETomek combines oversampling and undersampling strategies to boost the efficiency of the classifier model. The SMOTE method is used to oversample the minority group and then identify and eliminate samples from the majority group, as described in Tomek Links, to achieve a more equitable distribution of demographic characteristics. These methods outperform competing balancing methods when used to predict CVDs (*Sahid et al., 2022*).

## Feature reduction

**Feature extraction using PCA:** PCA is the most widely utilized unsupervised algorithm for feature extraction (*Uddin et al., 2021*). In our study, we use PCA to mitigate the dimensionality of the dataset. PCA converts a series of potentially correlated observations together into a collection of Principle Component (PC) values that are either linearly statistically independent observations (*Singh et al., 2017*). PCA produces eigenvectors in decreasing order, such as PC1, PC2, PC3, *etc*. Using the PCs, the analysis of the dataset can be much better than the original dataset. Moreover, it significantly decreases the algorithm's complexity while improving model performance in classification and regression.

**Feature selection using RF:** The feature selection process consists of choosing the most crucial features of a dataset and removing less important features that may help to improve the dataset's quality. Sometimes, the performance of the model increases by eliminating less significant features from the dataset (*Gayathri et al., 2022*). RF is one of the many techniques for feature selection. Due to its ensemble nature, RF is less prone to overfitting than other algorithms. Therefore, we employ it for feature selection. An RF is constructed by combining several DTs (*Oshiro, Perez & Baranauskas, 2012*). Internal nodes and leaves are unique to each tree. The internal node performs what to do to segment the data set into

two sets with strong responses according to the chosen feature. Internal node features are determined using some criterion. For regression tasks, variance reduction is used, while for classification, information gain and Gini impurity are used. When using information gain to train a DT, optimizing information gain yields the best split. Gini impurity indicates the impurity of a node, and its range is from 0 to 1. Moreover, lowering the value of the Gini impurity means a purer node, and we can calculate the average reduction in impurity for each feature. The feature gets more valuable as the nodes become purer. As well as, the feature significance is measured as the average of all trees in the forest. When the significance of all features is calculated, the much more significant features are chosen using a sequential backward elimination technique. Mathematically, Gini impurity of feature after each split $= 1 - \sum_{i=1}^{c} p_i^2$ with $Entropy = \sum_{i=1}^{c} p_i log_2 p_i$ and Information $Gain = 1 - Entropy$, where $c$ = number of classes of a dataset and $p_i$ = the probability of selecting random an element from class $i$.

## ML classifier

This study employs different supervised ML classifiers to analyze the heart disease prediction based on the properties of our preprocessed five different datasets in the inter-dataset discrepancy setup. A short description of each classifier is provided below.

**SVM:** SVM has become the most widespread supervised ML technique for classification and regression (*Rabbi et al., 2022*). In the SVM, perhaps every data item is depicted as just a point throughout $n$-dimensional space ($n$ represents the number of features). Hyperplanes, or decision boundaries, are used to help classify data items. Support vectors are the number of observations or trajectories nearest to the hyperplane which influences its position. The distance between observations or trajectories and the hyperplane is the margin. SVM's goal is to maximize this margin as much as feasible. The hyperplane with the biggest margin is the ideal hyperplane. To transform a low-dimensional input vector to a higher-dimensional vector, SVMs employ the kernel technique. The hyperplane is symbolized as $w.x + b = 0$, where $w$ is the hyperplane's normal vector and $b$ is indeed an offset. A decision rule must be defined to classify a point as negative or positive. The following is a definition of a decision rule:

$\vec{X} . \vec{w} - c \geq 0$
*putting* $- c$ *as* $b$, *we get*
$\vec{X} . \vec{w} + b \geq 0$ *hence*

$$f(x) = \begin{cases} +1, & \text{if } \vec{X}.\vec{w} + b \geq 0 \\ -1, & \text{if } \vec{X}.\vec{w} + b < 0 \end{cases}$$

If $\vec{X} . \vec{w} - c \geq 0$ is active, then the result is positive; otherwise, it shows a negative point. $w$ and $b$ are responsible for maximizing the margin distance. There are some problems with classifying using SVM that it does not perform very well when the dataset is large, and the dataset has more noise (*Maglogiannis et al., 2009*). Furthermore, if the target cases

overlap, or if the quantity of features for every piece of data surpasses the training data, SVM will perform poorly (*Hasan et al., 2023b*).

**RF:** RF is a DT-based bagging ensemble ML algorithm. It is used for classification and regression purposes (*Breiman, 2001*). A RF is an $m$-tree classifier composed of a set of tree-structured $h_1(x, Z_1), \ldots, h_n(x, Z_m)$, where the $Z_1, \ldots, Z_m$ are completely identical unbiased uniform random vectors. When given input $x$, each tree votes for the class with the highest support degree. A bootstrap sample of the data is used for its training set, and the data is then concatenated to generate a precise prediction. Each bootstrap sample is used as input for a DT, and the outputs from all trees are averaged to reach a consensus or mean. As a result, utilizing RF lessens the possibility of overfitting. Bootstrapping is training several separate DTs in parallel on different portions of the training dataset using different subsets of accessible characteristics (*Hasan, Islam & Sohag, 2023*). Bootstrapping minimizes the RF's overall variance (*Partopour, Paffenroth & Dixon, 2018*). The RF, on the other hand, is unsteady, which implies that a comparatively tiny variation in the information can significantly alter the effective DT's layout.

**MLP:** The MLP neural network consists of one input layer, several hidden layers, and an output layer (*Cinar, 2020*). An early step toward more complex NN was the representation of a single neuron called a perceptron. The depth represents the overall amount of layers, and the breadth represents the overall amount of a single layer. The backpropagation learning algorithm is used to train the neurons in the MLP. Neurons are arranged in layers. The amount of neurons present in the input layer for this pattern problem is proportional to the number of measurements. The amount of neurons within the output layer, on the other hand, is proportional to the number of classes. It is called feedforward since information is passed from $x$ to the function getting analyzed, then to the intermediate mathematical operations to define $f$, and consequently to the output $y$. Mathematically, $F(x) = y$.

**LR:** LR is an effective supervised ML technique for predicting the probability of a binary decision. Regularization is required in LR to avoid overfitting, especially if the number of training cases is limited or there are multiple parameters to understand (*Cawley & Talbot, 2010*). Furthermore, LR can classify multi-class classification. Binary data can be categorized using the logistic function as $\sigma(a) = \frac{1}{1-e^{-a}}$. In this case, $e$ is Euler's number, and $t$ is the input variable. When the value of $t$ is big enough, $(a) \to 1$ is returned, and when the value of $t$ is small, $(a) \to 0$ is returned. The logistic function is also known as the sigmoid function (*Sultan et al., 2023*). At the training stage, the logistic coefficients would be $a0, a1, a2\ldots.an$ for every instance $x1, x2, x3\ldots.xn$. The following formula updates the coefficients:

$$value = a_0 x_0 + a_1 x_1 + \cdots + a_n x_n$$
$$\sigma(t) = \frac{1}{1 - e^{-value}}$$
$$a = a + 1 * (y - \sigma(t)) * (1 - \sigma(t)) * \sigma(t) * x.$$

Each training instance's output value is $y$, and the value of the coefficient is initially 0. Where $x$ represents the unfair input for $b0$, always being 1, and $l$ represents the learning rate. The coefficient parameters are adjusted since the training stage correctly predicts the outcome. LR creates linear frontiers, and when the amount of observations is lower than

the number of features, overfitting may occur (*Babyak, 2004*). A major constraint of LR is the implication of linearity between independent and dependent variables.

**GNB:** The Bayes theorem serves as the basis for the supervised classification techniques in the naive Bayes family. The GNB variation is a naive Bayes variant depending upon that Gaussian normal distribution (*Wan et al., 2019*). The normal distribution is another name for the probability distribution. To predict future outcomes, GNB employs a probability distribution function. For every observation, the standard deviation and mean are first calculated. Probabilities are calculated depending on the mean and standard deviation of the test data, and a class level is assigned to each piece of data upon receipt. The probability of an event may change depending on the information available at the time (*Alwateer et al., 2021*), following Bayes' theorem. Because the naive Bayes classification model is easy to implement and would not require sophisticated incremental parameter estimation, it is ideal for large datasets.

**KNN:** KNN is an ML algorithm that labels unknown data points by considering $k$-nearest neighbors of that unknown data point (*Taneja et al., 2014*). More accurately, for each test data point, KNN considers only the first $k$-nearest data points of unknown data points in the training space. To find the nearest neighbor, KNN uses Euclidean distance. Let us consider unknown data points as $A(x_1, x_2, ....x_n)$ and others data point as $B(y_1, y_2, ....y_n)$. The distance between unknown data points and other data points can be defined using the Euclidean distance equation as follows:

$$d(A, B) = \sqrt{\sum_{i=1}^{n} (x_i - y_i)^2}.$$

KNN determines the distance between unknown data points and every other data point in this manner. Then, KNN selects $k$-nearest data points according to the minimum calculated distance. As KNN already finds $k$-nearest data points, now it figures out how many data points belong to each class label. Then voting is applied to identify the majority class label and assign this identified label to unknown data points. For a binary classification problem, choosing the value of $k$, an odd number, would be better to break the tie of two classes (*Elkan, 2011*). Roughly, if there are $m$ classes, then avoid the value of $k$ multiplied by $m$ to break the tie.

**XGB or XGBoost:** One well-liked boosting algorithm is gradient boosting. In gradient boosting, each predictor aims to fix the flaws of the one before it. Gradient Boosted DTs are implemented in XGBoost in sequential form. In the XGBoost algorithm, all independent variables are given weights, and also the DT that predicts outcomes utilizes these weights to predict the outcomes (*Nabipour et al., 2020*). Variables that the tree erroneously predicted are given a higher weight before being placed into the subsequent DT. Gradient-boosting DTs of this type are therefore assembled to develop a reliable and precise model.

**DT:** DT is a supervised ML classifier, $h : X \rightarrow Y$, which traverses from a tree's root node to a leaf to determine the label associated with instance $x$. The leaf node of DT contains a specific label. Input space partitioning determines the successor child at every node along the root-to-leaf path. Typically, a preset set of splitting rules or one of $x$'s attributes is the

basis for the splitting. DT uses the Gini index, which is a number that measures how accurately a split is between the groups that are categorized (*Hasan et al., 2023a*). A score between 0 and 1, where 1 represents a random distribution of the elements within classes, is evaluated using the Gini index.

## Performance measure metrics

In this section, we discuss the performance measurement techniques employed in this study. We utilize accuracy, precision, recall, F1 score, specificity, Cohen Kappa, AUC, MCC, NPV, and PPV. The details of the metrics are outlined below:

**Accuracy:** Accuracy is defined as the ratio of correctly classified instances to the total number of instances. One of the fundamental performance assessment techniques is accuracy. The ratio of (True Positive (*TP*) + True Negative (*TN*)) and (True Positive (*TP*) + *TN* + False Positive (*FP*) + False Negative (*FN*)) is accuracy. In imbalanced data, sometimes accuracy leads to false illusions. Mathematically,

$$Accuracy = \frac{TP + TN}{TP + TN + FP + FN}.$$

**Precision:** Precision indicates the accuracy of prediction of the positive instance by the model. In binary classification, the ratio between *TP* and the summation of *TP* and *FP* is the way to find the precision (*Tharwat, 2021*). Our goal with imbalanced data is to minimize *FP*. Precision also computes the accuracy of the minority class. As such, where *FP* is high, it is highly recommended to use precision as a performance measurement technique. Mathematically,

$$Precision = \frac{TP}{TP + FP}.$$

**Recall:** Recall indicates the number of times positive events have been positively represented out of all positive events. The recall is also referred to as True Positive Rate (*TPR*) or sensitivity. It is the ratio of the *TP* and the sum of *TP* and *FN*. To reduce the *FN* imbalance data, recall is appropriate to measure performance. Mathematically,

$$Recall = \frac{TP}{TP + FN}.$$

**F1-score:** The harmonic mean of precision and recall is defined as the F1-score. Only accuracy cannot judge whether a model is precise enough. When precision and recall are both high, a model makes more sense. To capture this important feature, the F1-score is calculated. It is frequently used to differentiate the effectiveness of two classifiers. Its range is [0,1], and the greater the F1-score value, the more reasonable the model is. Mathematically,

$$\text{F1score} = \frac{2 \times Precision \times Recall}{Precision + Recall}$$

**Table 6 Hyperparameters tuning of the classifiers using grid search CV.**

| Classifier | Parameter and value |
| --- | --- |
| SVM | *probability* = True, *C* = 10, *gamma* = 0.1, *kernel* = linear |
| KNN | *n_neighbors* = 5, *algorithm* = 'ball_tree', *weights* = 'distance', *metric* = 'minkowski', *p* = 2 |
| DT | *criterion* = 'gini', *splitter* = 'best' |
| LR | *penalty* = 'l2', *C* = 1.0, *random_state* = None, *solver*='lbfgs', *max_iter* = 100, *multi_class* = 'auto', *verbose* = 0 |
| RF | *n_estimators* = 100, *random_state* = 42 |
| XGB | *n_estimators* = 100, *booster* = 'gbtree', *gamma* = 0 |
| GNB | *priors* = None, *var_smoothing* = 1e−09 |
| MLP | *hidden_layer_sizes* = (100), *activation* = 'relu', *solver* = 'adam', *alpha* = 0.0001, *batch_size* = 'auto', *learning_rate* = 'constant' |

**Specificity:** The ratio between the *TN* and the sum of *TN* and *FN* is defined as specificity. Where negative classification is a high priority, specificity is used in that case. Mathematically,

$$\text{Specificity} = \frac{TN}{TN + FP}.$$

**Area under the curve (AUC):** A classifier's ability to distinguish between positive and negative classifications is called AUC (*Deepak & Ameer, 2019*). The AUC measures how efficiently the model differentiates between negative and positive classes. The higher the AUC, the better. Its value ranges from 0 to 1. When AUC = 1, the classifier can discriminate between all Positive and Negative class occurrences. If the AUC is 0, the classifier predicts that all negatives are positives and all positives are negatives. When AUC = 0.5, the classifier is unable to distinguish between Positive and Negative class instances.

**Cohen's Kappa:** Cohen's Kappa (CK) is a statistic that measures inter-rater agreement for categorical variables. When classifying data, agreement between expected and actual labels is assessed, taking into account the possibility of coincidental agreement. The formula for Cohen's Kappa is:

$$Cohen's\ Kappa = \frac{2 \times (TP \times TN - FN \times FP)}{(TP + FP) \times (FP + TN) + (TP + FN) \times (FN + TN)}.$$

**Matthews correlation coefficient (MCC):** MCC in classification evaluates the correlation between expected and actual binary classifications, taking into account TP, TN, FP, and FN. The formula for MCC is:

$$MCC = \frac{(TP \times TN - FP \times FN)}{(TP + FP) \times (FP + TN) \times (TP + FN) \times (FN + TN)}.$$

**Table 7 Performance of the classifiers on the single and the combined dataset without applying the proposed preprocessing pipeline.**

| Model | Accuracy | | Precision | | Recall | | F1 Score | | Specificity | | AUC | | CK | | MCC | | NPV | | PPV | |
| --- | --- | --- | --- | --- | --- | --- | --- | --- | --- | --- | --- | --- | --- | --- | --- | --- | --- | --- | --- | --- |
| Algo | Min | Max | Min | Max | Min | Max | Min | Max | Min | Max | Min | Max | Min | Max | Min | Max | Min | Max | Min | Max |
| SVM | 0.54 | 0.95 | 0.29 | 0.91 | 0.54 | 0.95 | 0.38 | 0.93 | 0.00 | 0.90 | 0.17 | 0.84 | 0.08 | 0.61 | 0.03 | 0.57 | 0.02 | 0.56 | 0.00 | 0.61 |
| KNN | 0.46 | 0.95 | 0.46 | 0.91 | 0.46 | 0.95 | 0.46 | 0.93 | 0.00 | 0.79 | 0.37 | 0.76 | 0.03 | 0.57 | 0.00 | 0.77 | 0.12 | 0.76 | 0.23 | 0.55 |
| DT | 0.67 | 0.92 | 0.67 | 0.90 | 0.67 | 0.92 | 0.67 | 0.90 | 0.00 | 0.90 | 0.46 | 0.88 | 0.11 | 0.83 | 0.15 | 0.87 | 0.12 | 0.72 | 0.17 | 0.64 |
| RF | 0.76 | 0.95 | 0.76 | 0.91 | 0.76 | 0.95 | 0.75 | 0.93 | 0.00 | 0.94 | 0.50 | 0.96 | 0.24 | 0.87 | 0.17 | 0.82 | 0.09 | 0.71 | 0.13 | 0.81 |
| XGB | 0.70 | 0.94 | 0.70 | 0.93 | 0.70 | 0.94 | 0.70 | 0.92 | 0.00 | 0.95 | 0.52 | 0.96 | 0.28 | 0.52 | 0.17 | 0.73 | 0.11 | 0.75 | 0.09 | 0.73 |
| GNB | 0.32 | 0.89 | 0.69 | 0.92 | 0.32 | 0.89 | 0.42 | 0.89 | 0.08 | 0.97 | 0.38 | 0.93 | 0.00 | 0.54 | 0.00 | 0.67 | 0.02 | 0.52 | 0.07 | 0.57 |
| LR | 0.67 | 0.94 | 0.67 | 0.91 | 0.67 | 0.94 | 0.67 | 0.92 | 0.00 | 0.96 | 0.36 | 0.95 | 0.16 | 0.89 | 0.11 | 0.87 | 0.03 | 0.88 | 0.02 | 0.84 |
| MLP | 0.51 | 0.95 | 0.62 | 0.91 | 0.51 | 0.95 | 0.48 | 0.93 | 0.00 | 0.96 | 0.17 | 0.92 | 0.01 | 0.88 | 0.11 | 0.69 | 0.04 | 0.64 | 0.05 | 0.73 |

**Negative predictive value (NPV):** NPV in classification represents the proportion of true negative predictions among all negative predictions. The equation to calculate NPV is as follows:

$$\text{NPV} = \frac{TN}{TN + FN}.$$

**Positive predictive value (PPV):** PPV in classification represents the proportion of true positive predictions among all positive predictions. The equation for PPV calculation is given below:

$$\text{PPV} = \frac{TP}{TP + FP}.$$

**Receiver operating characteristics (ROC):** ROC was developed for military radar operations, specifically to describe an operator's skill at determining whether or not an incoming radar signal was from a friendly or hostile aircraft. ROC curve helps to visualize, organize, and choose classifiers according to their performance. The *TPR* is depicted along the *y*-axis and the *FPR* along the *x*-axis, with each threshold represented by a separate point on the curve.

## RESULT AND ANALYSIS

We first consider the datasets without employing our proposed preprocessing pipeline to analyze the inter-data discrepancy issue in heart disease prediction. After that, we use each part of the preprocessing pipeline one at a time to see how each component works. The findings are summarized and shown graphically with scatter plots and ROC curves.

### Experimental setting

The associated hyperparameters in the algorithms are determined with GS Cross Validation (CV) (*Patil & Bhosale, 2021*). Once the models have been fitted to the datasets,

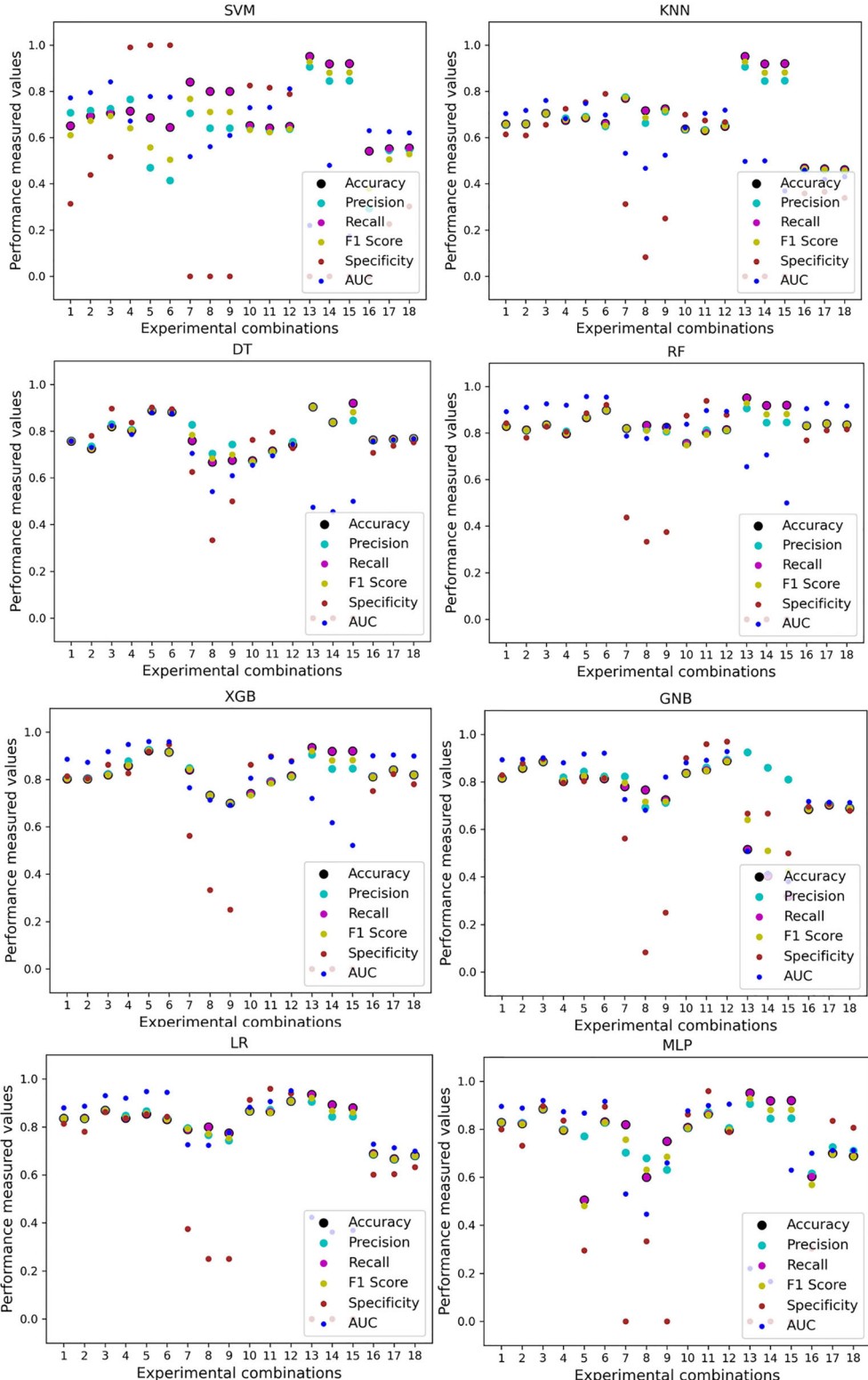

**Figure 3 Performance of the classifiers on individual and the combined dataset without applying the proposed preprocessing pipeline.** Scattered points indicate the low performances of the classifiers in different ratios of training and testing.

**Table 8** Performance of the classifiers on different combinations of the datasets without applying the proposed pipeline.

| Model | Accuracy | | Precision | | Recall | | F1 Score | | Specificity | | AUC | | CK | | MCC | | NPV | | PPV | |
|---|---|---|---|---|---|---|---|---|---|---|---|---|---|---|---|---|---|---|---|---|
| Algo | Min | Max | Min | Max | Min | Max | Min | Max | Min | Max | Min | Max | Min | Max | Min | Max | Min | Max | Min | Max |
| SVM | 0.07 | 0.70 | 0.04 | 0.89 | 0.07 | 0.70 | 0.08 | 0.74 | 0.00 | 0.92 | 0.23 | 0.76 | 0.03 | 0.68 | 0.07 | 0.84 | 0.12 | 0.90 | 0.13 | 0.88 |
| KNN | 0.07 | 0.67 | 0.06 | 0.93 | 0.07 | 0.67 | 0.06 | 0.67 | 0.03 | 0.88 | 0.01 | 0.68 | 0.07 | 0.66 | 0.06 | 0.89 | 0.10 | 0.89 | 0.07 | 0.90 |
| DT | 0.27 | 0.82 | 0.27 | 0.92 | 0.27 | 0.82 | 0.26 | 0.86 | 0.16 | 0.83 | 0.28 | 0.74 | 0.23 | 0.88 | 0.17 | 0.91 | 0.29 | 0.92 | 0.27 | 0.92 |
| RF | 0.19 | 0.81 | 0.16 | 0.93 | 0.19 | 0.81 | 0.18 | 0.85 | 0.11 | 0.96 | 0.10 | 0.87 | 0.17 | 0.88 | 0.15 | 0.88 | 0.17 | 0.90 | 0.18 | 0.93 |
| XGB | 0.22 | 0.81 | 0.14 | 0.93 | 0.22 | 0.81 | 0.17 | 0.86 | 0.10 | 0.91 | 0.07 | 0.83 | 0.23 | 0.80 | 0.17 | 0.94 | 0.22 | 0.89 | 0.20 | 0.88 |
| GNB | 0.23 | 0.81 | 0.23 | 0.92 | 0.23 | 0.81 | 0.23 | 0.81 | 0.24 | 0.97 | 0.14 | 0.85 | 0.13 | 0.78 | 0.15 | 0.80 | 0.22 | 0.89 | 0.17 | 0.86 |
| LR | 0.23 | 0.91 | 0.20 | 0.93 | 0.23 | 0.91 | 0.23 | 0.92 | 0.10 | 0.97 | 0.15 | 0.87 | 0.27 | 0.88 | 0.13 | 0.84 | 0.22 | 0.92 | 0.26 | 0.90 |
| MLP | 0.20 | 0.85 | 0.18 | 0.93 | 0.20 | 0.85 | 0.18 | 0.88 | 0.04 | 0.96 | 0.13 | 0.81 | 0.20 | 0.88 | 0.17 | 0.79 | 0.12 | 0.88 | 0.15 | 0.89 |

the algorithm's output can be evaluated. Moreover, all the hyperparameters and their respective values are provided in Table 6.

## Analysis without applying the proposed preprocessing pipeline for the inter-dataset performance discrepancy

Firstly, we analyze the five single and combined datasets created from the five different datasets without employing our proposed preprocessing pipeline. We train the ML models using an 80:20, 70:30, and 50:50 ratio of training and testing data. The performance of the classifiers is tabulated in Table 7. Moreover, among all the algorithms, SVM, KNN, RF, and MLP demonstrate a maximum accuracy of 95% with a recall of 95%. Additionally, the F1 score of SVM, KNN, RF, and MLP is 93%, which might indicate the high performance of the algorithms. However, considering the AUC score and the minimum scores of different performance measurement techniques achieved by the algorithms, the superiority of RF is more obvious than all. On the other hand, GNB shows a minimum accuracy of 32% and a maximum of 89%, but in some cases and combinations, it performs better than other classifiers. In terms of precision and AUC, XGB performs better than other algorithms. Nevertheless, the specificity (true negative) of KNN is lower than others. GNB also shows a high specificity as the accuracy is less than other algorithms in case of maximum accuracy. With these kinds of results, it is tough to find superior classifiers, and the performance of the algorithms is not stable in terms of precision and recall. In some cases, the classifiers show 0 specificity, which indicates that there were false positives, but no true negatives; all actual non-cases were incorrectly identified as positive. As such, having both of these suggests that everything was projected to be positive, whether it was an actual instance or not.

In Table 7, we only outline the minimum and maximum scores of different performance measurement techniques for individual algorithms. It is hard to find how the performance of the algorithms varies with the different ratios of training and testing the datasets. To show the variation in the performance of the algorithms, we plot the values and generate Fig. 3, where each subplot visualizes the performance of each algorithm. The scatter plot of

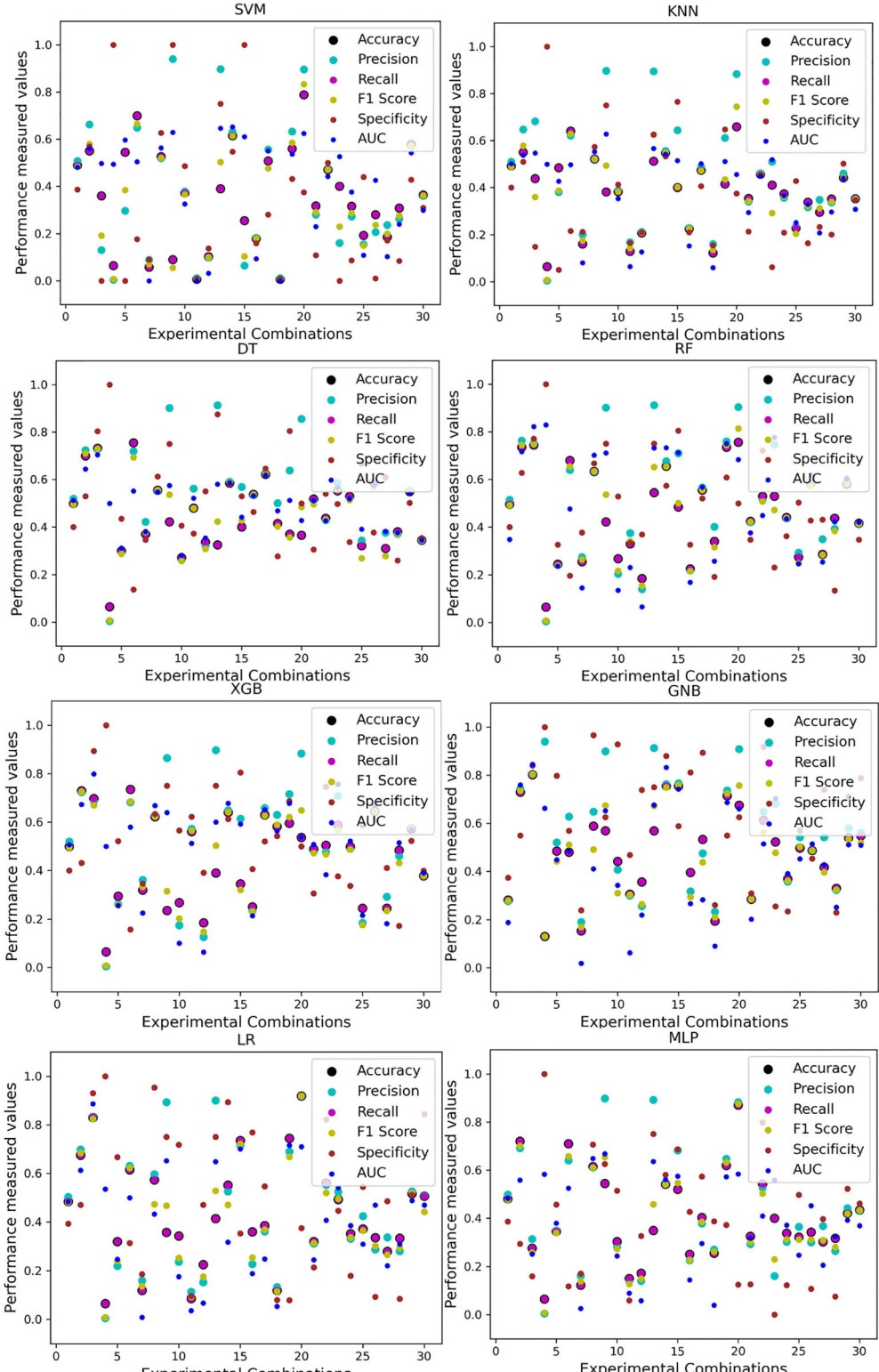

**Figure 4 Performance of the classifiers on different combinations of the dataset without applying the proposed preprocessing pipeline.** Scattered points indicate the low performances of the classifiers, which is the indicator of inter-dataset performance discrepancy.

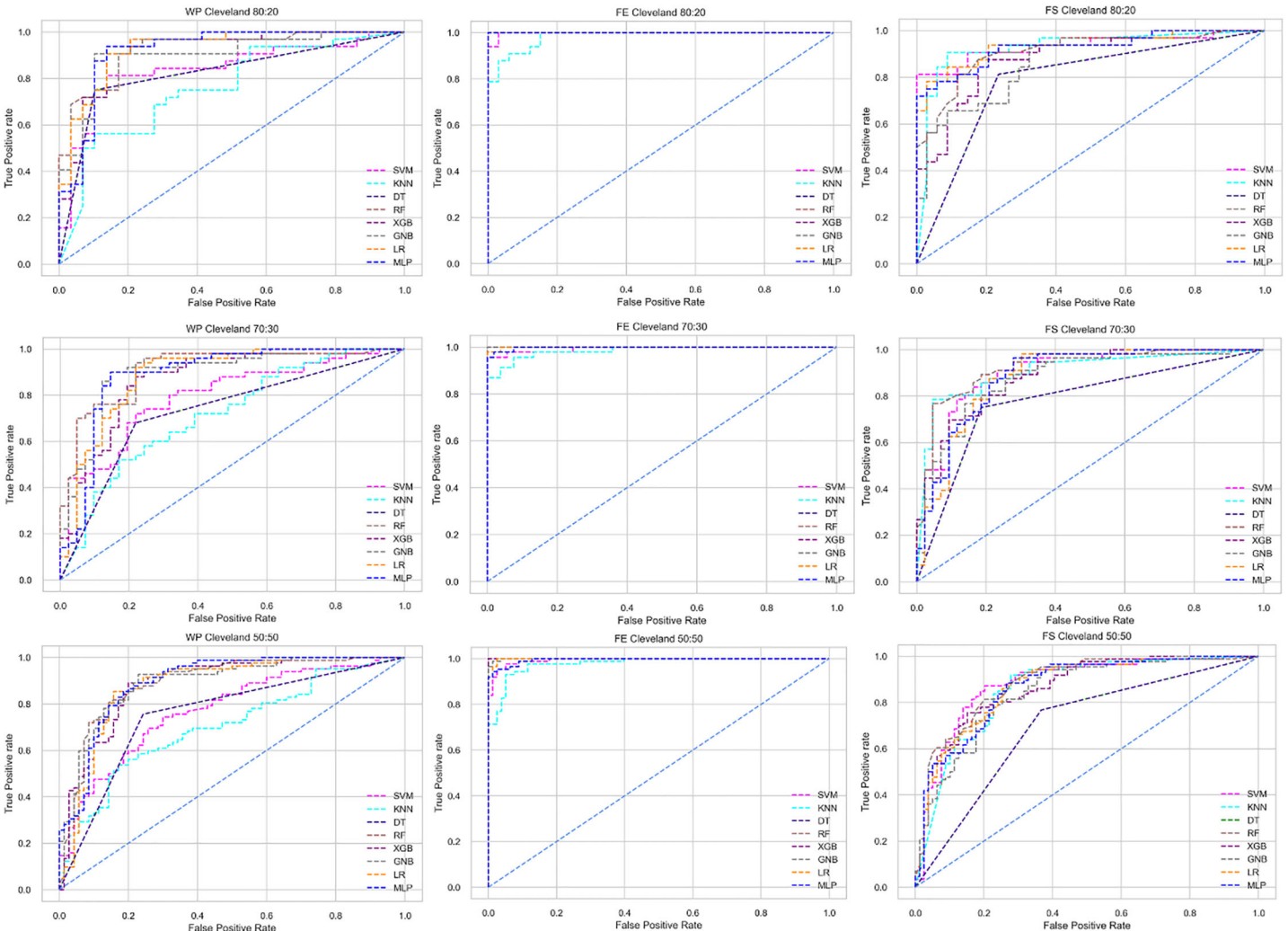

**Figure 5 ROC to show the performance of the classifiers for different ratios of training and testing in the Cleveland dataset.** The columns represent, without proposed preprocessing pipeline, PCA-used pipeline, and RF-used pipeline, respectively.

RF is less scattered than other algorithms. More scatter indicates a lower overall performance of the classifier. SVM, KNN, XGB, and MLP are more scattered than other classifiers, where Table 7 shows the performance of those algorithms is better than others. It is clear that the performance of the algorithms is better in some cases, but they could be more stable in predicting heart disease for different datasets.

For the Switzerland and Long Beach VA datasets, most of the classifier shows poor performance because these two datasets have many missing values and many inconsistencies and outliers. When we go through the inter-dataset setup, we use the datasets in different combinations for training and testing, which shows a disaster in the results of the classifiers in Table 8. The performance goes down to 7% in minimum accuracy for SVM and KNN. In this setup, LR shows a maximum 91% accuracy with 93%

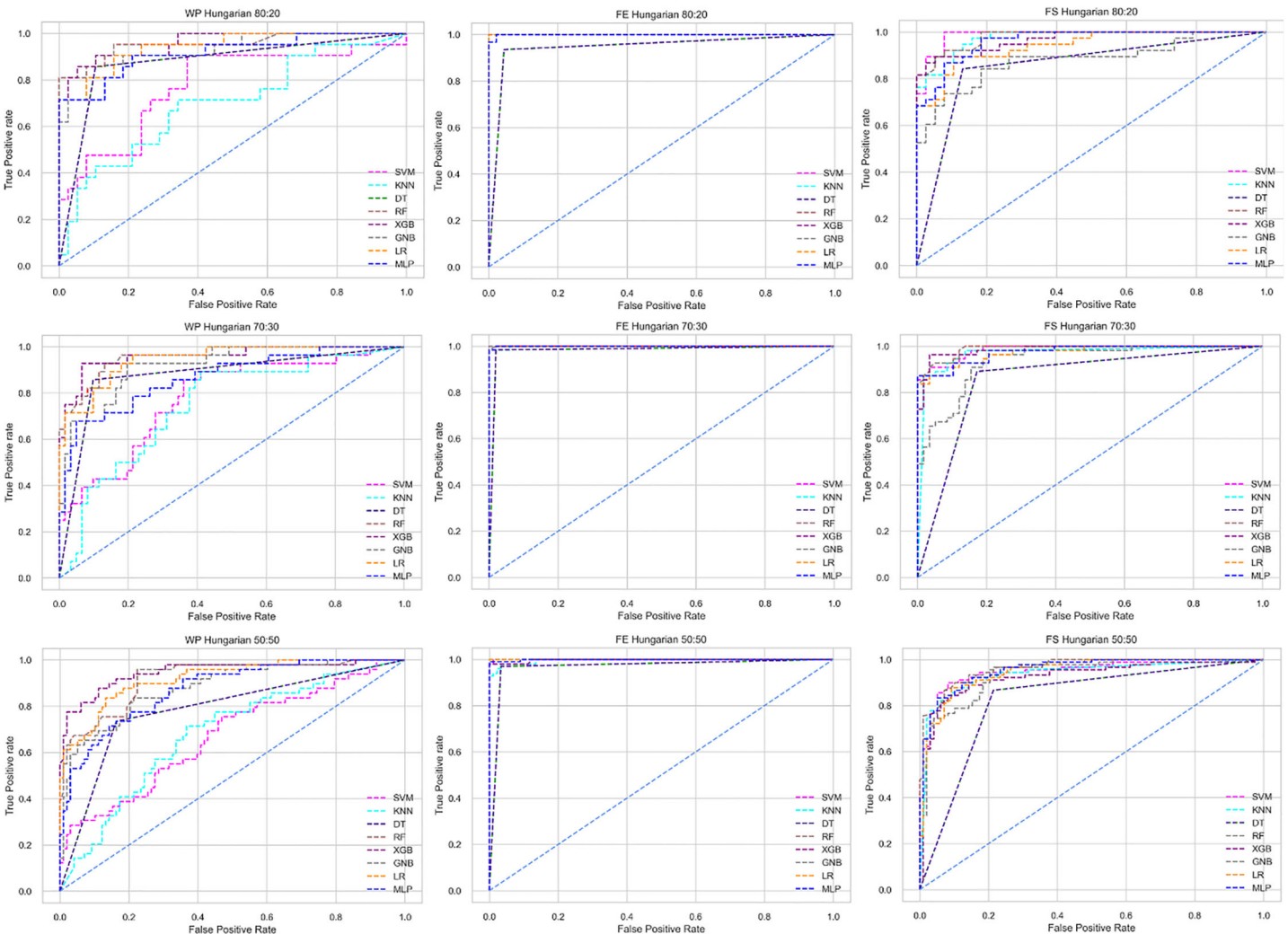

**Figure 6 ROC to show the performance of the classifiers for different ratios of training and testing in the Hungarian dataset.** The columns represent, without proposed preprocessing pipeline, PCA-used pipeline, and RF-used pipeline, respectively.

precision, 91% recall, 97% specificity, and 87% AUC, score. MLP also shows 85% accuracy and recall, 93% precision, 96% specificity, 81% AUC, 89% CK, 87% MCC, 88% NPV, and 84% PPV score. DT, RF, XGB, and GNB show more than 80% accuracy and recall and more than 90% precision. Among these four classifiers, the RF and GNB have good specificities 96%, and 97% respectively. From this table, we cannot clearly define the classifier's performance. To demonstrate the algorithm's performance more clearly, we plot all of the performance measure technique values of different combinations in Fig. 4. This figure shows the classifiers' performance individually in different subplots. We get a clear indication of the classifier's performance. However, the values are more scattered, which is proof of an inter-dataset performance discrepancy in heart disease prediction.

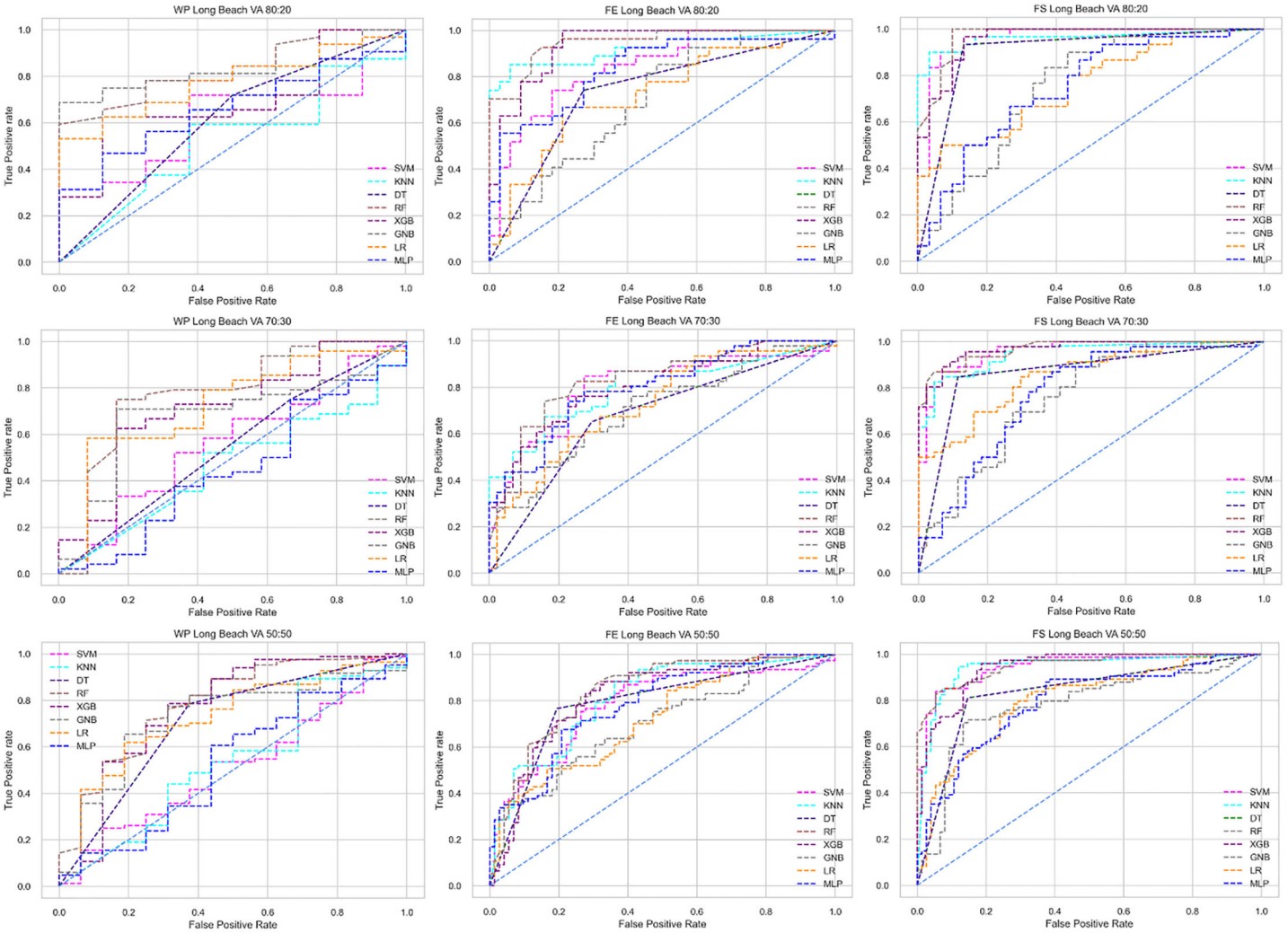

**Figure 7** **ROC to show the performance of the classifiers for different ratios of training and testing in the Long Beach VA dataset.** The columns represent, without proposed preprocessing pipeline, PCA-used pipeline, and RF-used pipeline, respectively.

The ROC curve also shows the performance of the classifiers for three different stages (without pipeline, PCA equipped, RF equipped) of different datasets individually. In Figs. 5–10 shows different ROC curves of different datasets in different states.

## Analysis of applying PCA to mitigate the inter-dataset performance discrepancy

We use a preprocessing pipeline to extract features using PCA from the datasets to minimize the inter-dataset performance discrepancy. In terms of the individual and combined datasets, the performance of the classifiers improves compared to the previous analysis, where we did not use our proposed preprocessing pipeline. The results of the classifiers for individual datasets are in Table 9. In this table, GNB achieves a minimum

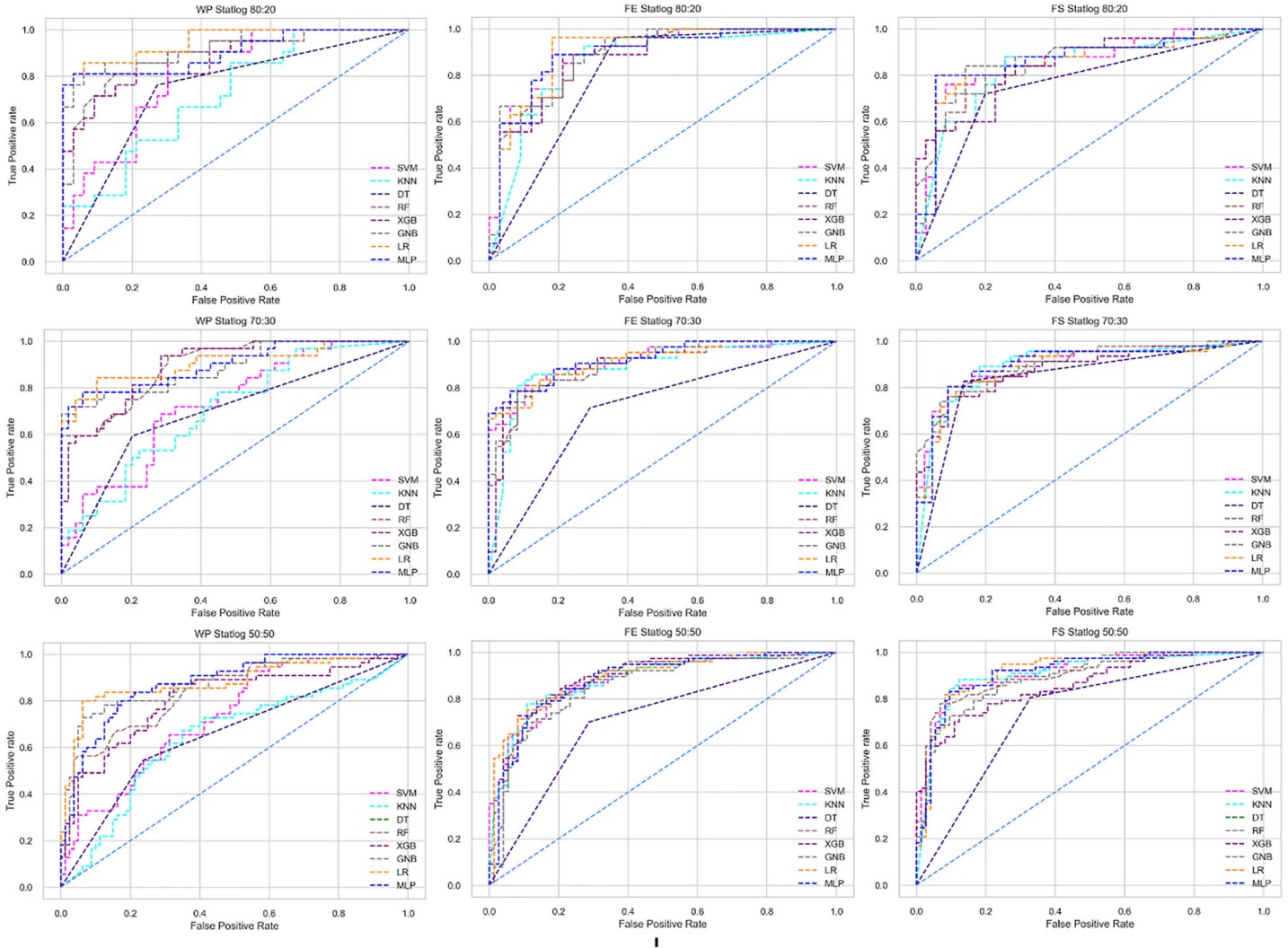

**Figure 8 ROC to show the performance of the classifiers for different ratios of training and testing in the Statlog dataset.** The columns represent, without proposed preprocessing pipeline, PCA-used pipeline, and RF-used pipeline, respectively.

accuracy of 60% compared to 32% in Table 7 for pipeline data without preprocessing. In Table 7 the maximum accuracy is 95%, but after using the preprocessing pipeline, we get 100% accuracy, precision, recall, and an F1 score for SVM, DT, RF, XGB, and GNB. In addition, KNN and LR show 99% accuracy, precision, recall, and F1-score, while MLP shows 98% accuracy, recall, F1 score, and 99% precision. All the classifiers show 100% specificity and an AUC score. In addition, the classifiers' minimum score is higher than the result in Table 7. All the classifier's results vary within the minimum and maximum ranges that are tabulated in Table 9. All the performance measure values are visualized in Fig. 11 by some scatter plots, where each subplot represents the scores for individual classifiers. The figure clearly shows that the results are not as dispersed as in Fig. 3. It indicates that our proposed methodology improves the classifier's performance at a significant level.

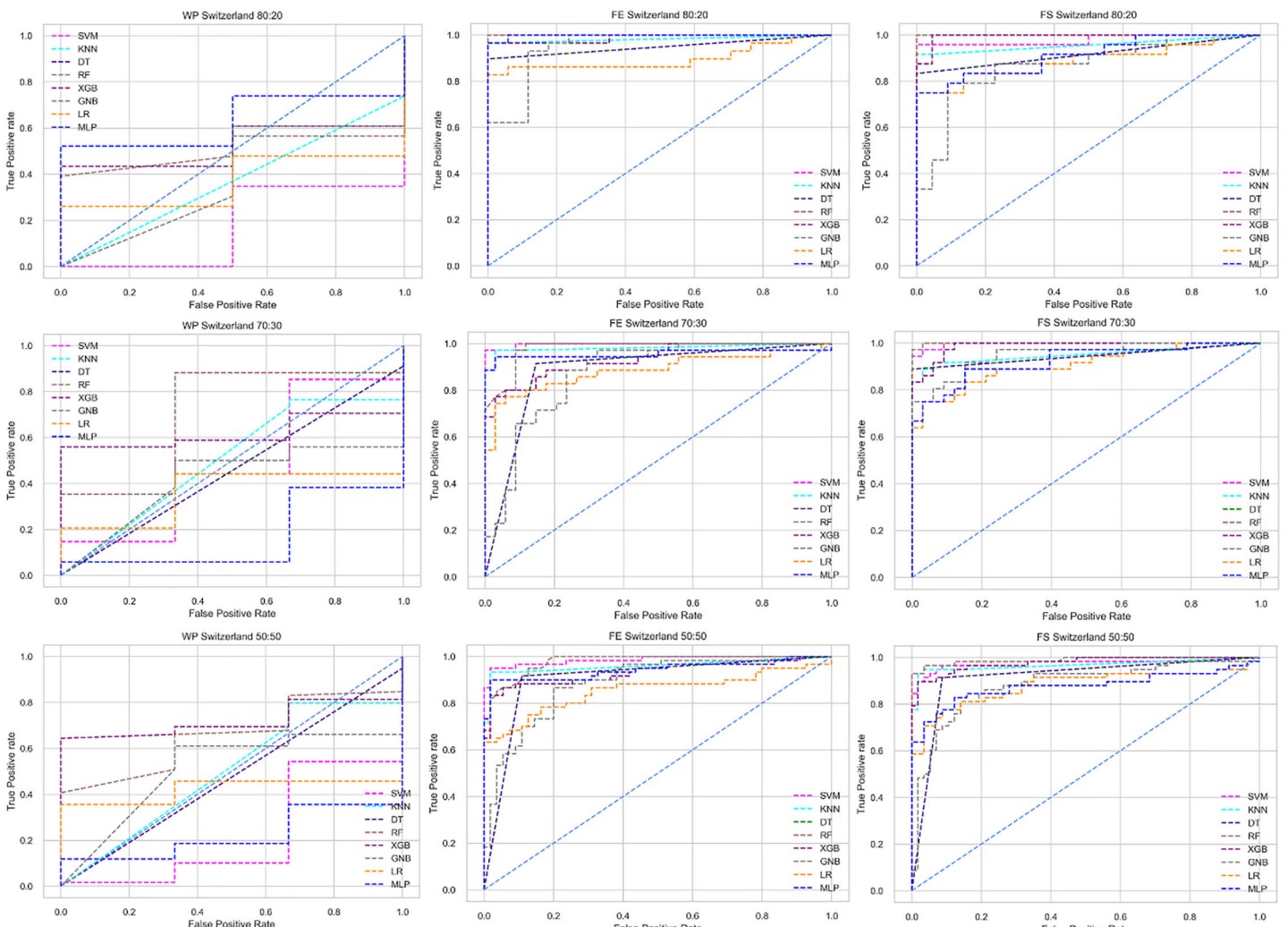

**Figure 9  ROC to show the classifiers' performance for different training and testing ratios in the Switzerland dataset.** The columns represent, without proposed preprocessing pipeline, PCA-used pipeline, and RF-used pipeline, respectively.

As with the single and combined datasets, we use the same preprocessing pipeline in the different combinations of the dataset that is presented in Table 5. Table 8 and Fig. 4 show the performance of the classifiers and indicate that the inter-data discrepancy occurred here. However, after applying PCA, we mitigated the discrepancy issue and improved the accuracy significantly. In Table 10, GNB achieved 57% accuracy, the lowest among the algorithms, whereas, in Table 5, the minimum accuracy was only 7% by SVM and KNN. On the other hand, the maximum accuracy for the dataset that did not go through the proposed preprocessing pipeline was 91% by LR. After applying the PCA, we get 92% accuracy by RF. Though the highest range of accuracy change is insignificant, the data discrepancy removal is significant. In Fig. 12, the subplots are not too scattered like in the previous Fig. 4, which proves the removal of data discrepancy of the inter-dataset setup. If we compare Table 10 with the previous Table 5, we get a clear indication that PCA

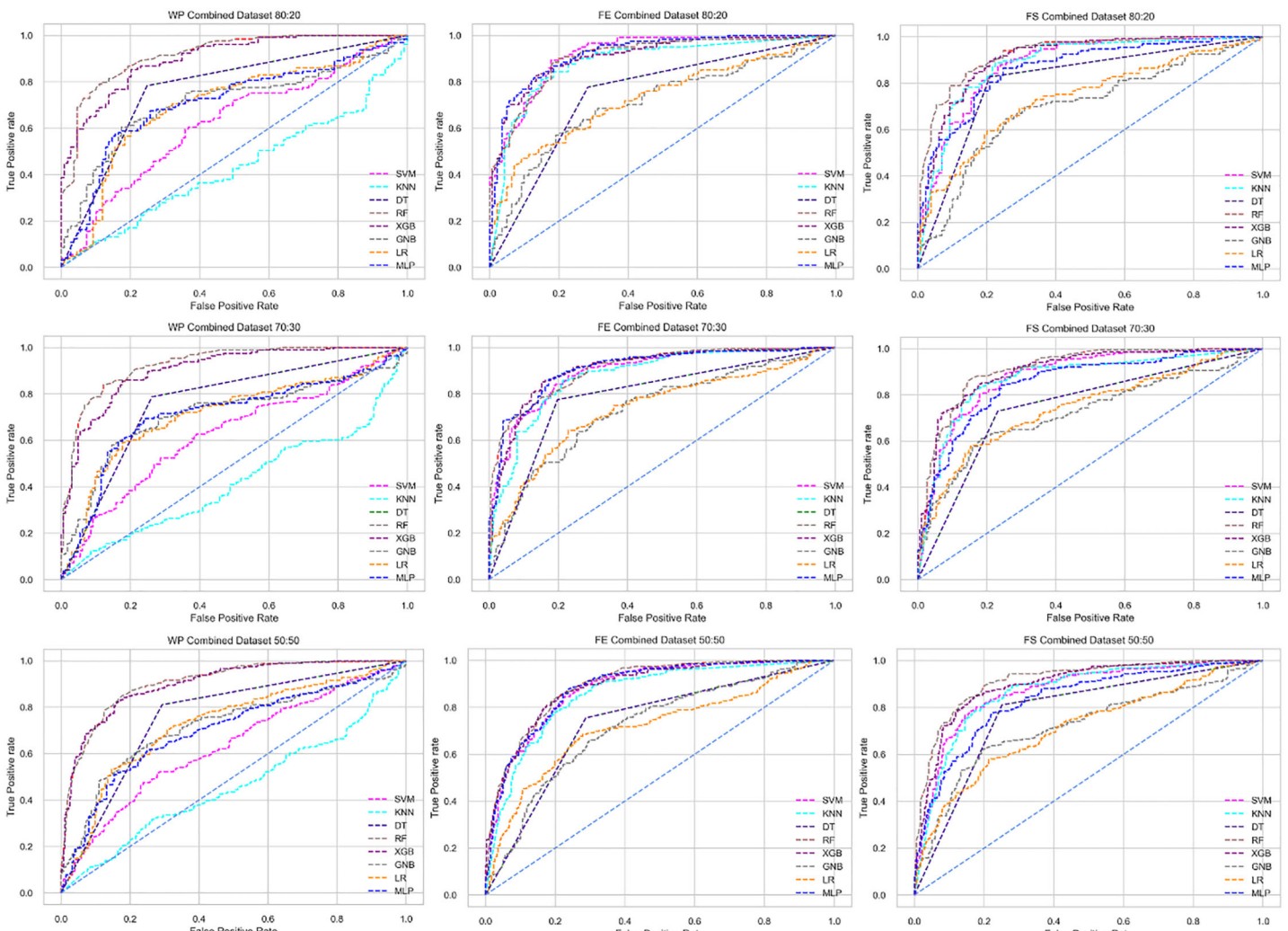

**Figure 10 ROC to show the classifiers' performance for different training and testing ratios in the Combined dataset.** The columns represent, without proposed preprocessing pipeline, PCA-used pipeline, and RF-used pipeline, respectively.

**Table 9 Performance of the classifiers on the single and the combined dataset applying PCA on the proposed pipeline.**

| Model | Accuracy | | Precision | | Recall | | F1 Score | | Specificity | | AUC | | CK | | MCC | | NPV | | PPV | |
|---|---|---|---|---|---|---|---|---|---|---|---|---|---|---|---|---|---|---|---|---|
| Algo | Min | Max | Min | Max | Min | Max | Min | Max | Min | Max | Min | Max | Min | Max | Min | Max | Min | Max | Min | Max |
| SVM | 0.70 | 0.99 | 0.70 | 1.00 | 0.70 | 1.00 | 0.70 | 1.00 | 0.70 | 1.00 | 0.78 | 1.00 | 0.70 | 0.99 | 0.69 | 0.99 | 0.70 | 0.99 | 0.72 | 0.98 |
| KNN | 0.69 | 0.99 | 0.73 | 0.99 | 0.67 | 0.99 | 0.68 | 0.99 | 0.76 | 1.00 | 0.80 | 1.00 | 0.67 | 0.98 | 0.69 | 0.99 | 0.70 | 0.98 | 0.70 | 0.98 |
| DT | 0.68 | 0.99 | 0.68 | 1.00 | 0.68 | 1.00 | 0.68 | 1.00 | 0.64 | 1.00 | 0.68 | 1.00 | 0.67 | 0.98 | 0.69 | 0.98 | 0.66 | 0.97 | 0.68 | 0.97 |
| RF | 0.75 | 1.00 | 0.75 | 1.00 | 0.75 | 1.00 | 0.75 | 1.00 | 0.76 | 1.00 | 0.83 | 1.00 | 0.75 | 0.99 | 0.75 | 0.99 | 0.74 | 0.98 | 0.76 | 0.99 |
| XGB | 0.75 | 0.99 | 0.75 | 1.00 | 0.75 | 1.00 | 0.75 | 1.00 | 0.75 | 1.00 | 0.80 | 1.00 | 0.75 | 0.98 | 0.75 | 0.99 | 0.75 | 0.98 | 0.75 | 0.98 |
| GNB | 0.60 | 0.99 | 0.60 | 1.00 | 0.60 | 1.00 | 0.60 | 1.00 | 0.61 | 1.00 | 0.70 | 1.00 | 0.62 | 0.96 | 0.60 | 0.99 | 0.62 | 0.97 | 0.64 | 0.98 |
| LR | 0.62 | 0.99 | 0.62 | 0.99 | 0.62 | 0.99 | 0.62 | 0.99 | 0.57 | 1.00 | 0.72 | 1.00 | 0.64 | 0.97 | 0.62 | 0.89 | 0.62 | 0.99 | 0.62 | 0.98 |
| MLP | 0.72 | 0.98 | 0.72 | 0.99 | 0.72 | 0.98 | 0.72 | 0.98 | 0.73 | 1.00 | 0.78 | 1.00 | 0.71 | 0.98 | 0.72 | 0.97 | 0.72 | 0.97 | 0.72 | 0.98 |

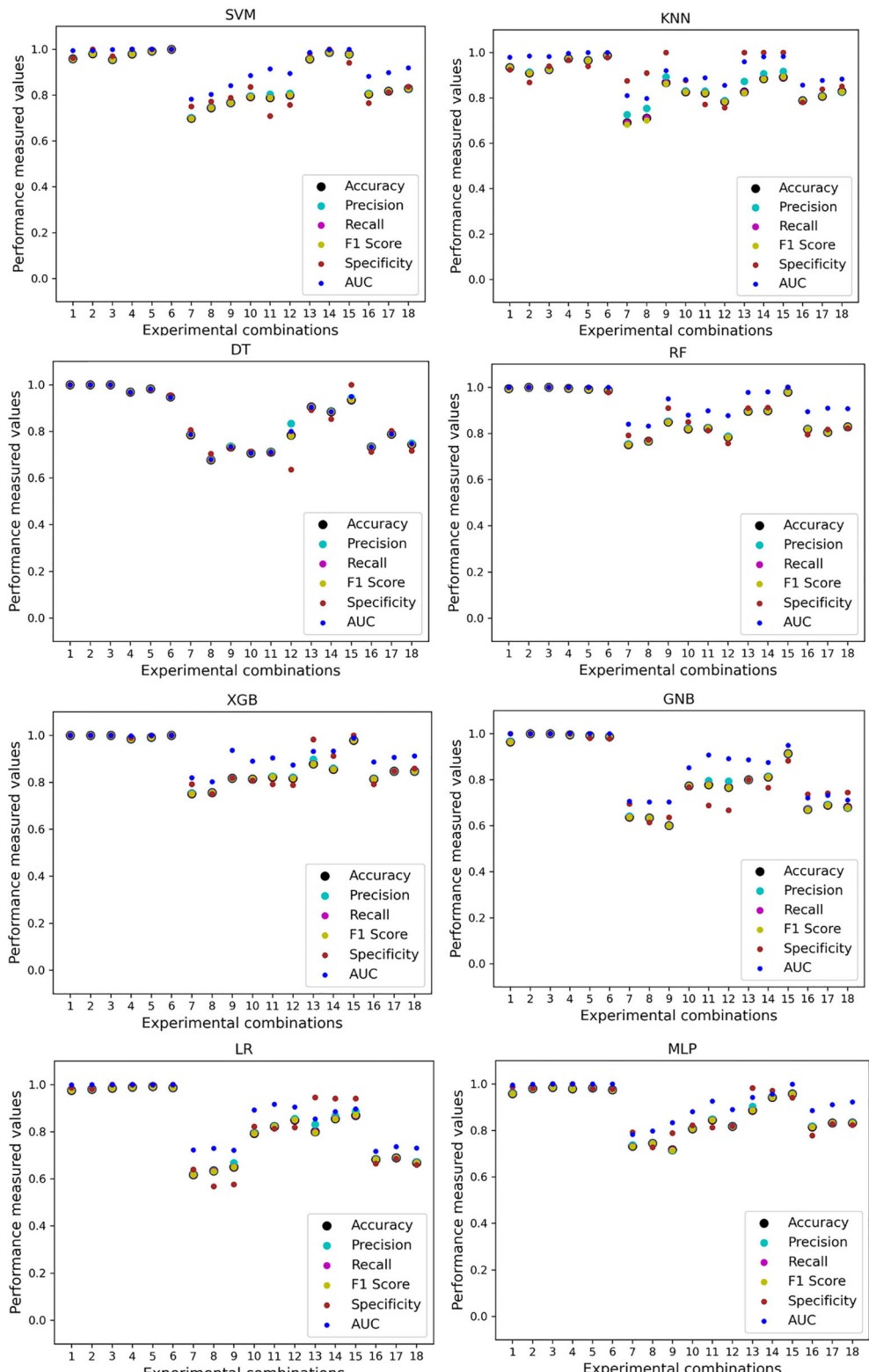

**Figure 11 Performance of the classifiers on single and combined dataset applying PCA as feature extraction techniques.** The scattered points indicate that the classifiers' performances are satisfactory, indicating the inter-dataset performance discrepancy minimization.

**Table 10 Performance of the classifiers on different combinations of the datasets applying PCA on the proposed pipeline.**

| Model | Accuracy | | Precision | | Recall | | F1 Score | | Specificity | | AUC | | CK | | MCC | | NPV | | PPV | |
|---|---|---|---|---|---|---|---|---|---|---|---|---|---|---|---|---|---|---|---|---|
| **Algo** | **Min** | **Max** | **Min** | **Max** | **Min** | **Max** | **Min** | **Max** | **Min** | **Max** | **Min** | **Max** | **Min** | **Max** | **Min** | **Max** | **Min** | **Max** | **Min** | **Max** |
| SVM | 0.77 | 0.89 | 0.77 | 0.89 | 0.77 | 0.89 | 0.77 | 0.89 | 0.74 | 0.91 | 0.85 | 0.96 | 0.53 | 0.78 | 0.52 | 0.84 | 0.76 | 0.93 | 0.74 | 0.91 |
| KNN | 0.74 | 0.89 | 0.74 | 0.89 | 0.74 | 0.89 | 0.74 | 0.89 | 0.71 | 0.95 | 0.84 | 0.96 | 0.47 | 0.79 | 0.50 | 0.77 | 0.72 | 0.88 | 0.76 | 0.90 |
| DT | 0.72 | 0.84 | 0.73 | 0.84 | 0.72 | 0.84 | 0.72 | 0.84 | 0.68 | 0.88 | 0.72 | 0.85 | 0.45 | 0.69 | 0.35 | 0.72 | 0.67 | 0.95 | 0.69 | 0.87 |
| RF | 0.80 | 0.92 | 0.80 | 0.92 | 0.80 | 0.92 | 0.80 | 0.92 | 0.76 | 0.96 | 0.88 | 0.97 | 0.60 | 0.84 | 0.59 | 0.83 | 0.80 | 0.98 | 0.77 | 0.89 |
| XGB | 0.78 | 0.90 | 0.79 | 0.91 | 0.78 | 0.90 | 0.78 | 0.90 | 0.74 | 0.93 | 0.87 | 0.97 | 0.57 | 0.81 | 0.48 | 0.98 | 0.73 | 0.98 | 0.76 | 0.90 |
| GNB | 0.57 | 0.89 | 0.59 | 0.89 | 0.57 | 0.89 | 0.56 | 0.89 | 0.65 | 0.91 | 0.65 | 0.94 | 0.16 | 0.78 | 0.28 | 0.73 | 0.58 | 0.91 | 0.65 | 0.83 |
| LR | 0.60 | 0.87 | 0.61 | 0.87 | 0.60 | 0.87 | 0.60 | 0.87 | 0.60 | 0.88 | 0.63 | 0.92 | 0.20 | 0.73 | 0.21 | 0.66 | 0.57 | 0.86 | 0.61 | 0.81 |
| MLP | 0.78 | 0.89 | 0.78 | 0.89 | 0.78 | 0.89 | 0.78 | 0.89 | 0.73 | 0.90 | 0.85 | 0.95 | 0.55 | 0.95 | 0.51 | 0.84 | 0.75 | 0.93 | 0.74 | 0.91 |

improves the overall performance of the classifiers. The scores are unstable in Table 5, and performances vary in different combinations. However, in Table 10, all the classifiers show stable performance with good accuracy. Here RF shows the superiority of prediction over other classifiers. All the second columns of each graph show the performance of the algorithms, and we get a clear indication that the performance of the PCA-based preprocessing pipeline helps to mitigate the data discrepancy.

## Analysis of applying RF to mitigate the inter-dataset performance discrepancy

Like PCA, we employ RF to select the features in the preprocessing pipeline to mitigate the inter-data performance discrepancy problem in heart disease prediction. We use this process in the single and combined dataset analyses and the different combinations of the dataset analyses. "Analysis of Applying PCA to Mitigate the Inter-Dataset Performance Discrepancy" shows the analysis results on the dataset that does not undergo our proposed preprocessing pipeline. The results of the classifiers for the individual and combined datasets and different combinations of the datasets are not satisfactory in this medical domain. Also, we show the result of PCA enabling the pipeline in "Analysis of Applying PCA to Mitigate the Inter-Dataset Performance Discrepancy", show the classifier's good performance, and prove the inter-data performance discrepancy mitigation. In this section, the results obtained by the classifiers for the single and combined datasets are in Table 11, and the performance of the classifiers for the different ratios of training testing is visualized in Fig. 13.

Table 11 only displays the lowest and highest results of different performance measurement approaches for specific algorithms. Concerning all algorithms, SVM and RF show 98% maximum accuracy, precision, recall, and an F1 score; specificity and an AUC score are both 100%, which is the highest value among all algorithms. Regarding the minimum accuracy score for accuracy, precision, recall, and an F1 score metric, RF also outperforms all other algorithms. Compared to Table 9, it is clear that SVM and RF show the highest value of 100% maximum accuracy, precision, recall, and an F1 score using

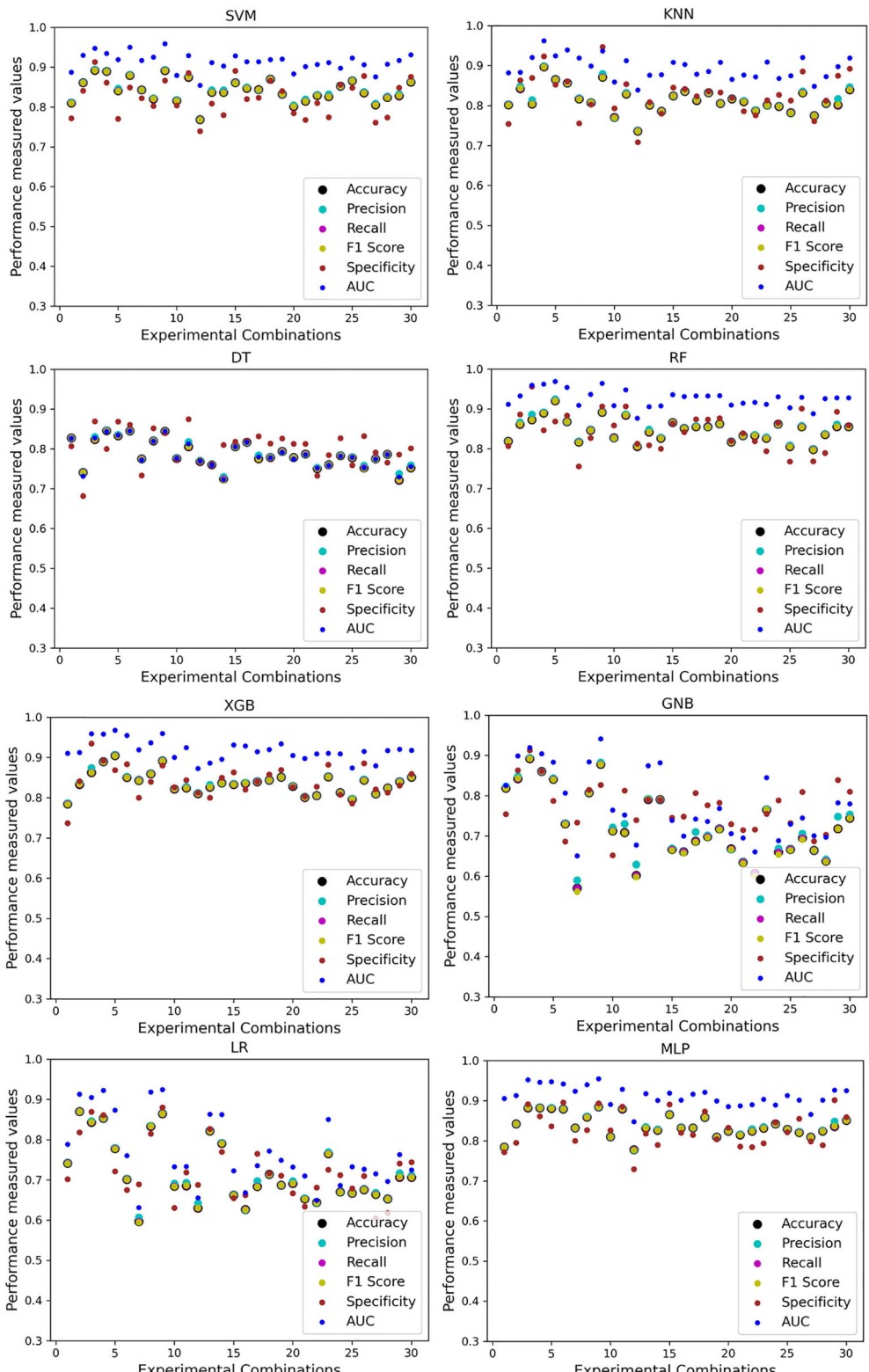

**Figure 12 Performance of the classifiers on different combinations of the dataset applying PCA as feature extraction techniques.** The scattered points show that the classifiers' performances are satisfactory, indicating the inter-dataset performance discrepancy minimization.

**Table 11 Performance of the classifiers on the single and the combined dataset applying RF on the proposed pipeline.**

| Model | Accuracy | | Precision | | Recall | | F1 Score | | Specificity | | AUC | | CK | | MCC | | NPV | | PPV | |
|-------|-----|-----|-----|-----|-----|-----|-----|-----|-----|-----|-----|-----|-----|-----|-----|-----|-----|-----|-----|-----|
| **Algo** | **Min** | **Max** | **Min** | **Max** | **Min** | **Max** | **Min** | **Max** | **Min** | **Max** | **Min** | **Max** | **Min** | **Max** | **Min** | **Max** | **Min** | **Max** | **Min** | **Max** |
| SVM | 0.77 | 0.98 | 0.78 | 0.98 | 0.77 | 0.98 | 0.77 | 0.98 | 0.72 | 1.00 | 0.86 | 1.00 | 0.76 | 0.97 | 0.80 | 0.96 | 0.77 | 0.98 | 0.76 | 0.99 |
| KNN | 0.78 | 0.93 | 0.79 | 0.94 | 0.78 | 0.93 | 0.78 | 0.93 | 0.73 | 1.00 | 0.83 | 0.98 | 0.78 | 0.97 | 0.77 | 0.96 | 0.77 | 0.98 | 0.78 | 0.98 |
| DT | 0.70 | 0.94 | 0.70 | 0.95 | 0.70 | 0.94 | 0.70 | 0.94 | 0.63 | 1.00 | 0.70 | 0.94 | 0.72 | 0.93 | 0.78 | 0.96 | 0.75 | 0.92 | 0.77 | 0.97 |
| RF | 0.80 | 0.98 | 0.80 | 0.98 | 0.80 | 0.98 | 0.80 | 0.98 | 0.72 | 1.00 | 0.85 | 1.00 | 0.82 | 0.98 | 0.80 | 0.96 | 0.83 | 0.98 | 0.88 | 0.99 |
| XGB | 0.75 | 0.95 | 0.76 | 0.95 | 0.75 | 0.95 | 0.75 | 0.95 | 0.68 | 0.97 | 0.85 | 0.99 | 0.79 | 0.98 | 0.82 | 0.95 | 0.79 | 0.96 | 0.80 | 0.98 |
| GNB | 0.69 | 0.87 | 0.69 | 0.87 | 0.69 | 0.87 | 0.69 | 0.87 | 0.57 | 0.88 | 0.70 | 0.95 | 0.66 | 0.90 | 0.68 | 0.86 | 0.67 | 0.88 | 0.70 | 0.86 |
| LR | 0.63 | 0.89 | 0.64 | 0.89 | 0.63 | 0.89 | 0.63 | 0.89 | 0.57 | 0.91 | 0.71 | 0.97 | 0.70 | 0.91 | 0.68 | 0.86 | 0.65 | 0.88 | 0.79 | 0.88 |
| MLP | 0.65 | 0.88 | 0.65 | 0.89 | 0.65 | 0.88 | 0.65 | 0.88 | 0.60 | 0.92 | 0.95 | 0.98 | 0.70 | 0.90 | 0.68 | 0.89 | 0.67 | 0.87 | 0.66 | 0.89 |

PCA-based FE. GNB algorithms perform poorly compared to other algorithms in terms of accuracy, precision, recall, F1 score, and specificity. On the other hand, GNB shows 100% maximum accuracy using PCA-based FE. XGB shows the second-highest performance, which is 95% maximum accuracy, precision, recall, and an F1 score; 97% specificity, and 99% AUC. Compared to Table 9, which is PCA-based FE, and Table 11, which is RF-based FS, we can conclude that all the algorithms show better performance in the case of PCA-based FE. The subplots in Fig. 13 are not as scattered as in Fig. 4, but they are a little more scattered than in Fig. 12, indicating a significant improvement in the classifier's performance.

As with the single and combined datasets, we employ the same preprocessing pipeline in the numerous different dataset combinations shown in Table 12. Table 10 and Fig. 4 show the classifiers' performance, which proves that the inter-data discrepancy has arisen. With the help of RF-based FS, the discrepancy issue is mitigated at a moderate level, improving the performance significantly higher than PCA-based FE for some classifiers. The RF and XGB algorithms show 96% maximum accuracy, precision, recall, and an F1 score; the AUC score is closed at 100%. KNN shows 94% maximum accuracy, precision, recall, and an F1 score. KNN also shows 97% maximum specificity, which is the highest among all other algorithms. DT achieves 95% accuracy, precision, recall, F1 score, specificity, and AUC score. These results show that RF, XGB, KNN, and DT algorithms improve performance significantly more than PCA-based FE using different combinations of datasets. In the case of SVM, the performance is nearly identical to that of the PCA-based FE result. GNB, LR, and MLP algorithms perform better in PCA-based FE than in RF-based FS. Figure 14 shows that classification performance for some algorithms improves more when using RF-based FS rather than PCA-based FE.

We use ROC to show the performance of the classifiers for checking the inter-dataset performance discrepancy among the five individual datasets. In Fig.15, we use one dataset for training and one for testing. The performance of the classifiers without preprocessing pipeline stages is unstable, and accuracy is below average in most cases. It indicates the inter-dataset performance discrepancy. After applying the PCA in the preprocessing

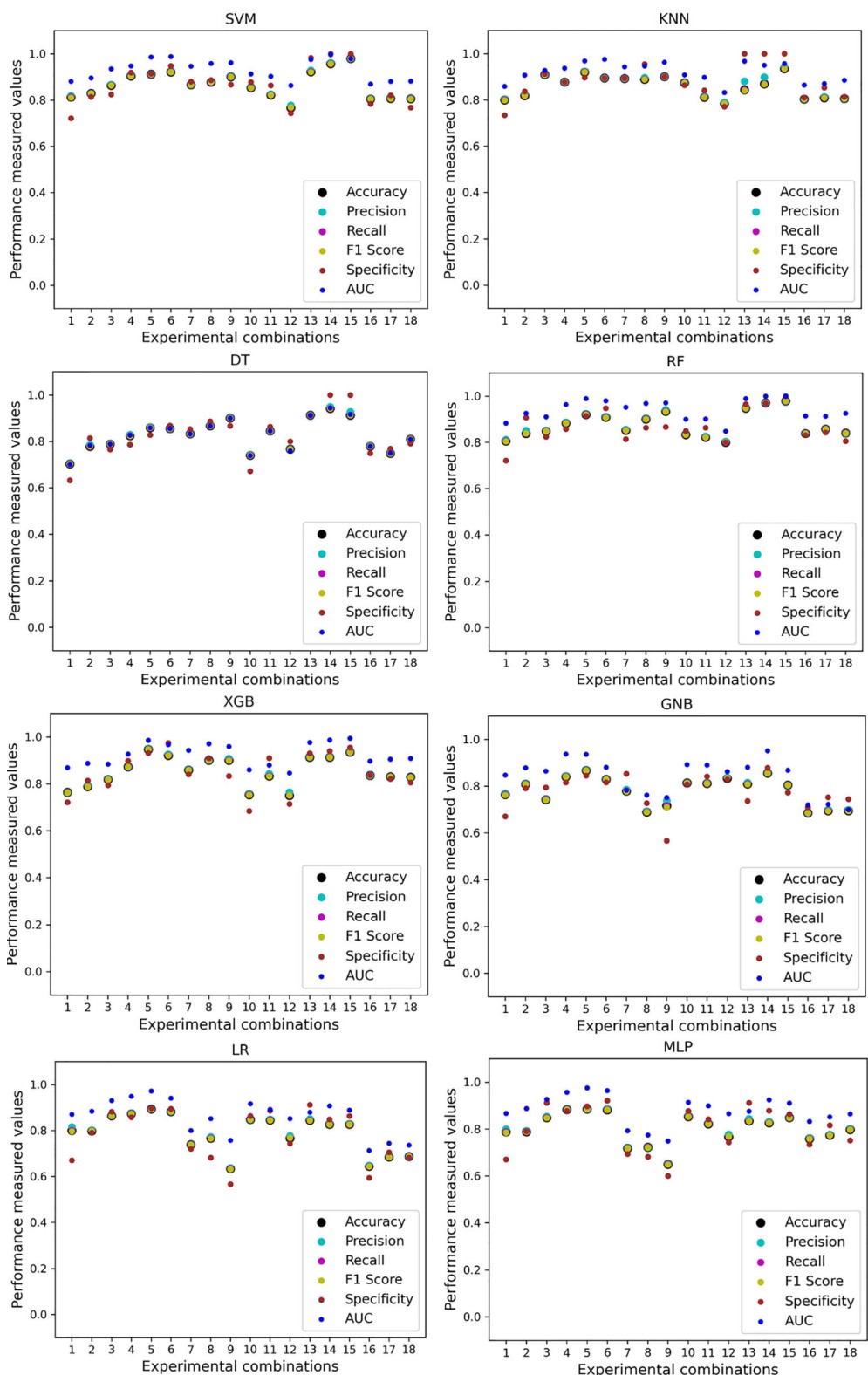

**Figure 13 Performance of the classifiers on single and combined dataset applying RF as feature selection techniques.** The scattered points show that the classifiers' performances are satisfactory, indicating the inter-dataset performance discrepancy minimization.

**Table 12 Performance of the classifiers on different combinations of the datasets applying RF.**

| Model | Accuracy | | Precision | | Recall | | F1 Score | | Specificity | | AUC | | CK | | MCC | | NPV | | PPV | |
|-------|-----|-----|-----|-----|-----|-----|-----|-----|-----|-----|-----|-----|-----|-----|-----|-----|-----|-----|-----|-----|
| Algo | Min | Max | Min | Max | Min | Max | Min | Max | Min | Max | Min | Max | Min | Max | Min | Max | Min | Max | Min | Max |
| SVM | 0.76 | 0.88 | 0.76 | 0.89 | 0.76 | 0.88 | 0.76 | 0.88 | 0.76 | 0.90 | 0.84 | 0.95 | 0.52 | 0.76 | 0.47 | 0.79 | 0.71 | 0.93 | 0.73 | 0.91 |
| KNN | 0.71 | 0.94 | 0.71 | 0.94 | 0.71 | 0.94 | 0.71 | 0.94 | 0.64 | 0.97 | 0.81 | 0.97 | 0.40 | 0.87 | 0.45 | 0.78 | 0.7 | 0.88 | 0.73 | 0.90 |
| DT | 0.67 | 0.95 | 0.67 | 0.95 | 0.67 | 0.95 | 0.67 | 0.95 | 0.60 | 0.95 | 0.66 | 0.95 | 0.33 | 0.91 | 0.38 | 0.76 | 0.67 | 0.86 | 0.71 | 0.90 |
| RF | 0.74 | 0.96 | 0.75 | 0.96 | 0.74 | 0.96 | 0.74 | 0.96 | 0.75 | 0.95 | 0.83 | 0.99 | 0.48 | 0.93 | 0.57 | 0.84 | 0.79 | 0.96 | 0.75 | 0.92 |
| XGB | 0.72 | 0.96 | 0.73 | 0.96 | 0.72 | 0.96 | 0.73 | 0.96 | 0.72 | 0.95 | 0.80 | 0.98 | 0.44 | 0.91 | 0.56 | 0.89 | 0.75 | 0.96 | 0.77 | 0.93 |
| GNB | 0.57 | 0.86 | 0.60 | 0.86 | 0.57 | 0.86 | 0.55 | 0.86 | 0.61 | 0.83 | 0.60 | 0.91 | 0.15 | 0.72 | 0.23 | 0.71 | 0.57 | 0.91 | 0.62 | 0.80 |
| LR | 0.63 | 0.83 | 0.64 | 0.84 | 0.63 | 0.83 | 0.63 | 0.83 | 0.56 | 0.86 | 0.63 | 0.92 | 0.27 | 0.67 | 0.22 | 0.68 | 0.58 | 0.89 | 0.57 | 0.83 |
| MLP | 0.71 | 0.86 | 0.72 | 0.86 | 0.71 | 0.86 | 0.71 | 0.86 | 0.64 | 0.86 | 0.80 | 0.93 | 0.43 | 0.72 | 0.54 | 0.93 | 0.72 | 0.91 | 0.73 | 0.89 |

pipeline, the performance of the classifiers is increased. It demonstrates that PCA feature extraction techniques can reduce the inter-dataset performance discrepancy. Also, the last columns show the ROC of the RF feature selection pipeline.

The results obtained by training the models using two datasets and testing only one dataset are visualized by the ROC curves in Fig. 16. The combination of 6 to 10 of Table 5 is here for the three phases. However, the performance of the classifiers is too low, and in most cases, it falls below the average line indicated in the first column subplots. After applying PCA and RF in the preprocessing phase, the performance of the classifiers improves dramatically, indicating that this proposed preprocessing pipeline minimizes the inter-dataset performance discrepancy problem. However, the performance of the classifiers is high in both PCA and RF, but compared to these two, RF feature selection shows better results in these cases. In conclusion, RF as a classifier is superior in predicting heart disease, and our proposed preprocessing pipeline mitigates the inter-dataset performance discrepancy problem in this domain.

The ROC curves of Fig. 17 are generated from the combinations of 11 to 15 of Table 5 where three datasets are in the training phase and 1 in the testing phase. Like the previous two phases, the classifiers' performance is too low when we don't apply the proposed preprocessing pipeline. Also, the performance is more scattered, and most cases are below 50% in accuracy. After applying the PCA and RF, we can recover these issues. The performance of the classifiers increases in a good position, and in most cases, it shows more than 85% accuracy. More specifically, the performance of the classifiers is better in the RF-integrated pipelines than in PCA-integrated pipelines.

When four datasets are in the training phases and one in the testing phases, the inter-dataset performance discrepancy occurs. Classifiers show a low performance that is not tolerable in this sensitive domain. We also apply our proposed preprocessing pipeline separately, where PCA and RF are mainly responsible for the dimensionality reduction of the datasets we already discussed. The performance of the classifiers increases after applying the preprocessing pipeline. The ROC is in Fig. 18.

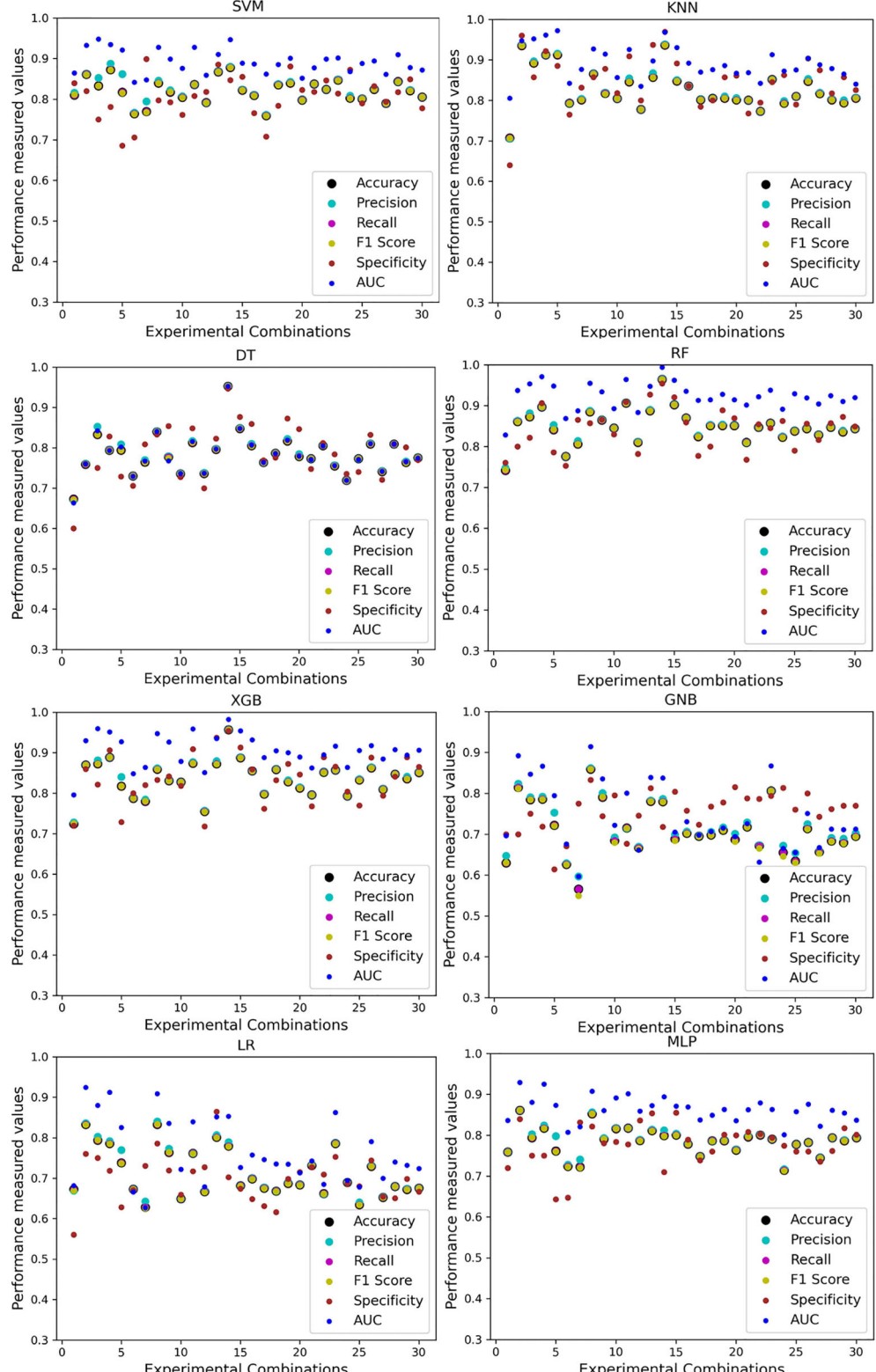

**Figure 14** **Performance of the classifiers on different combinations of datasets applying RF as feature selection techniques.** The scattered points show that the classifiers' performances are satisfactory, indicating the inter-dataset performance discrepancy minimization.

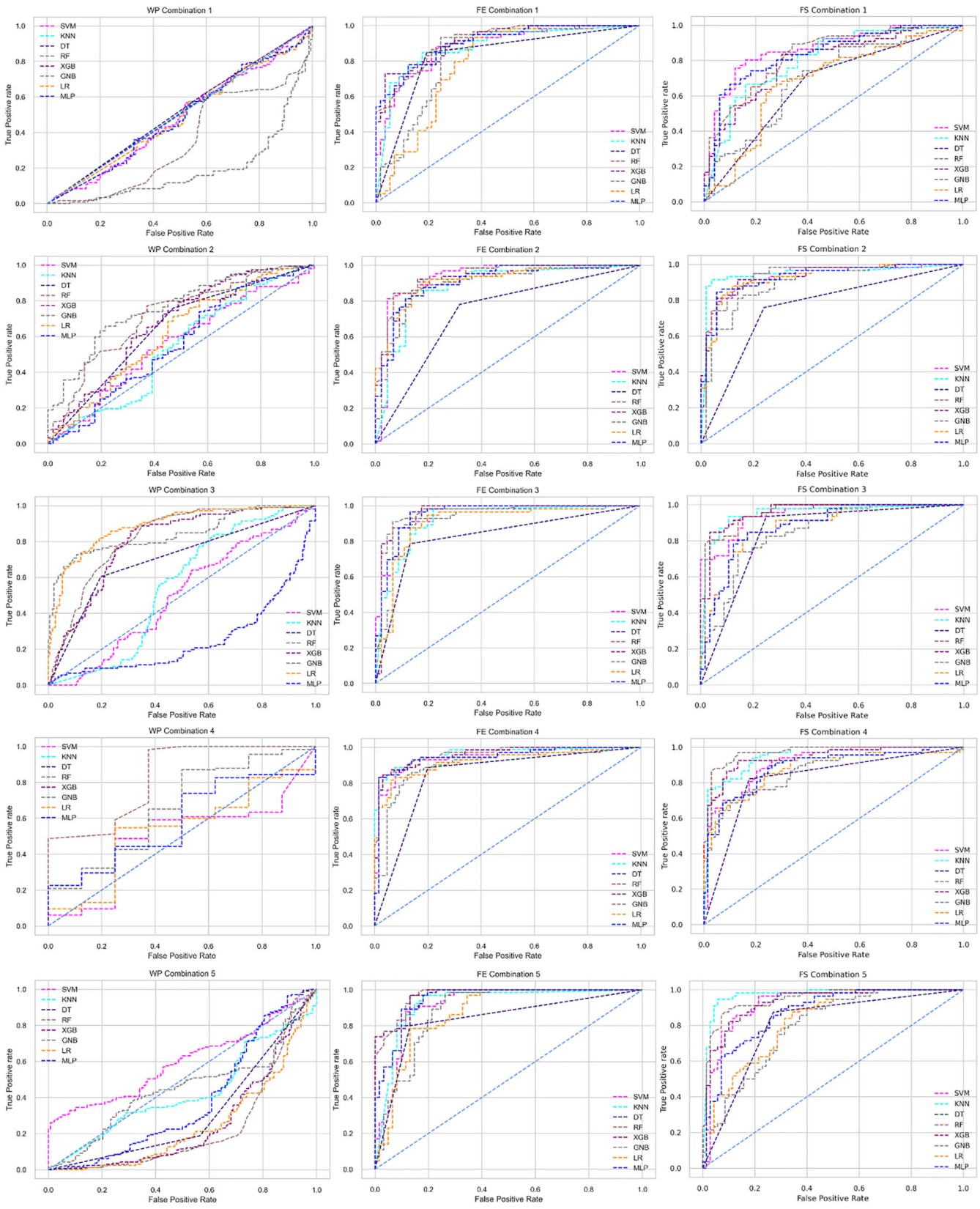

**Figure 15  ROC of the combination 1 (training) by 1 (testing).**

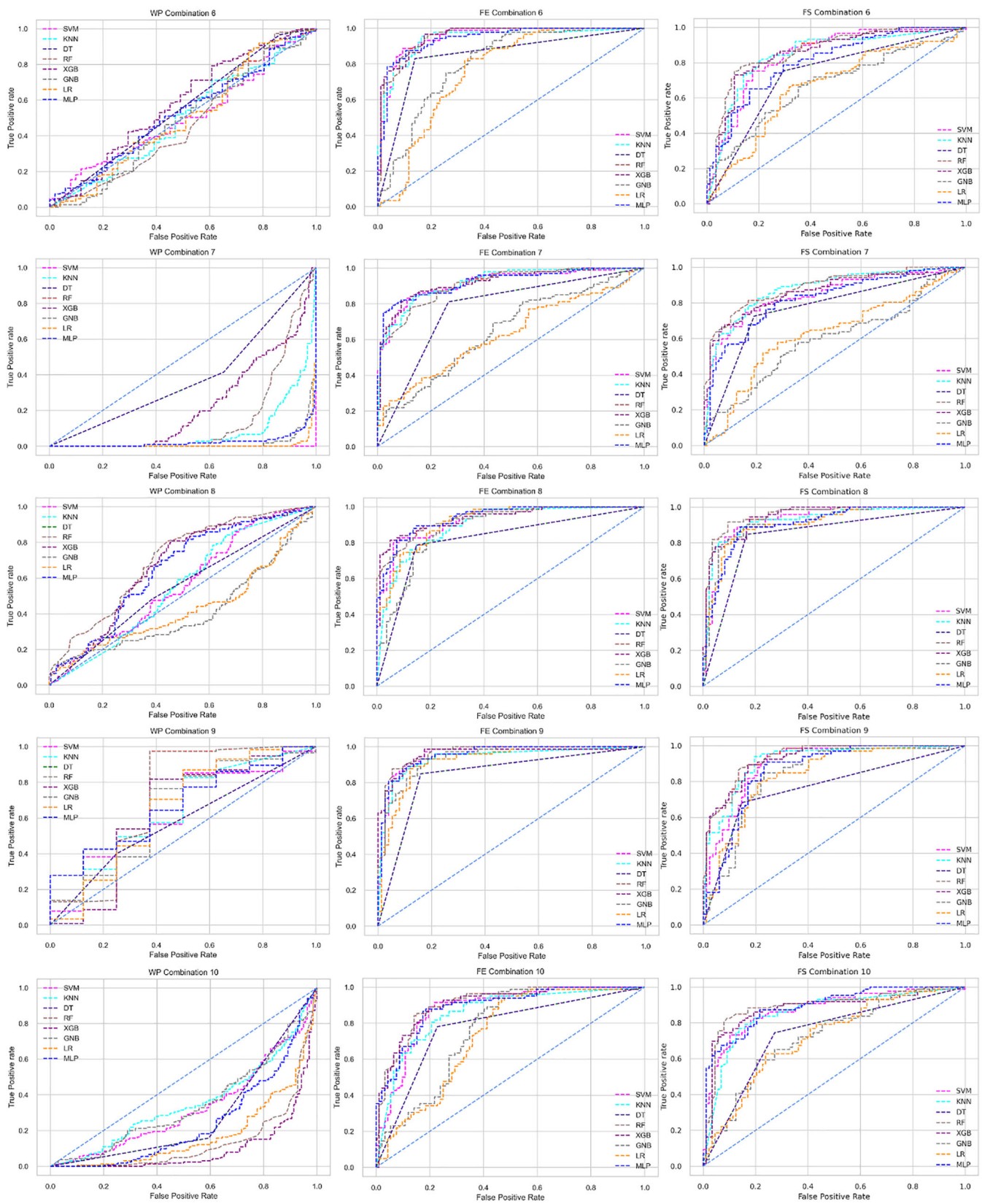

**Figure 16 ROC of the combination 2 (training) by 1 (testing).**

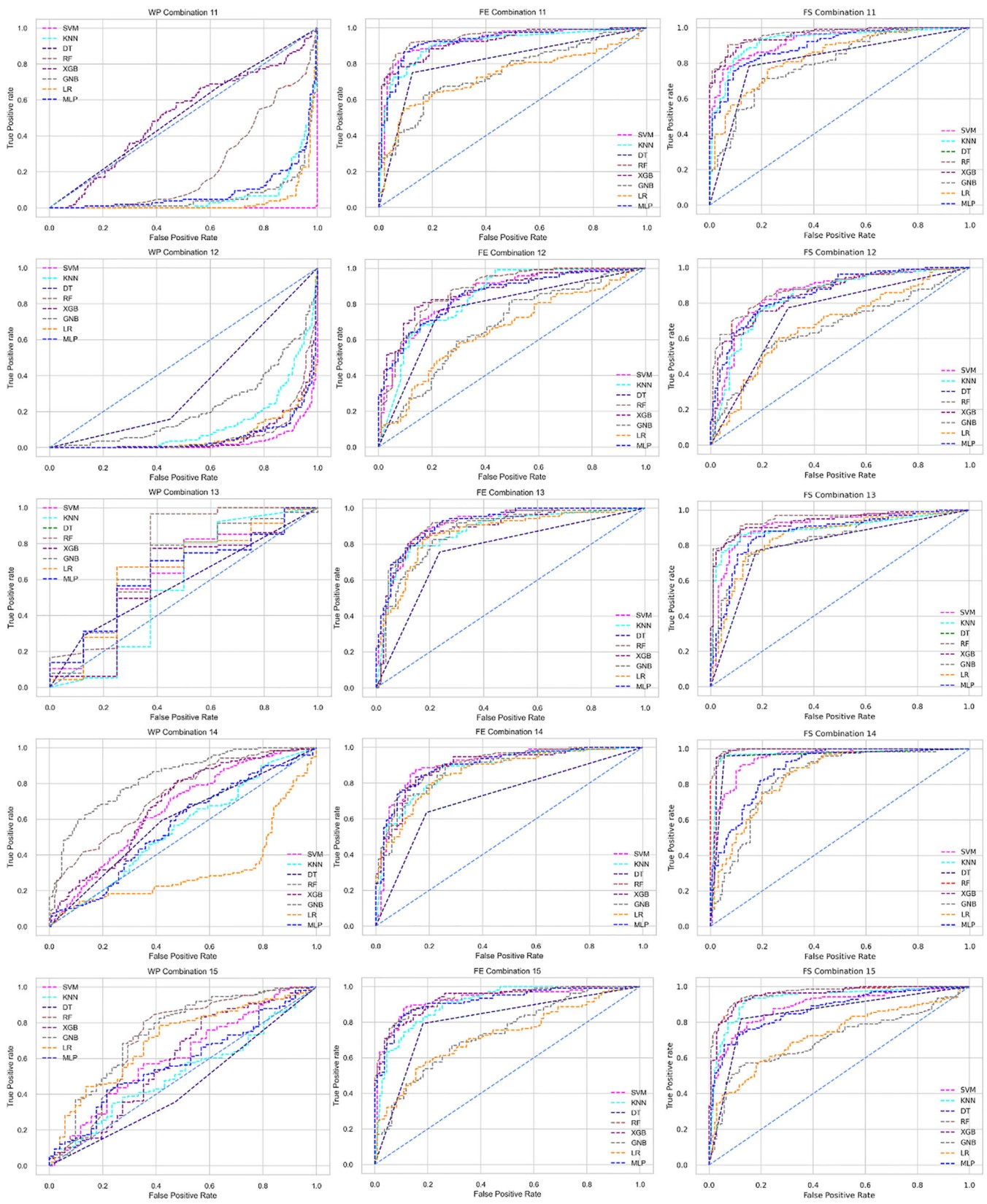

**Figure 17** ROC of the combination 3 (training) by 1 (testing).

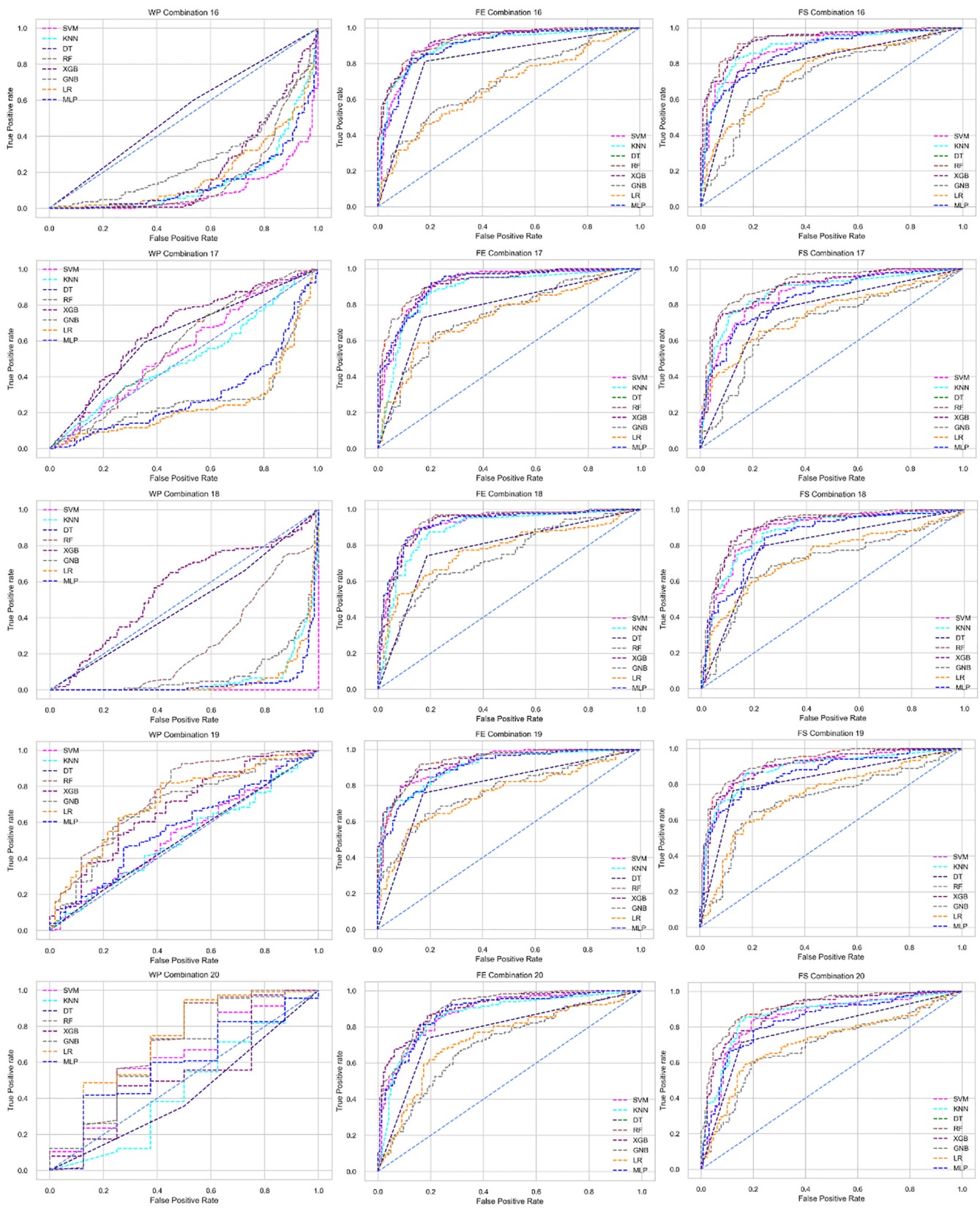

**Figure 18 ROC of the combination 4 (training) by 1 (testing).**

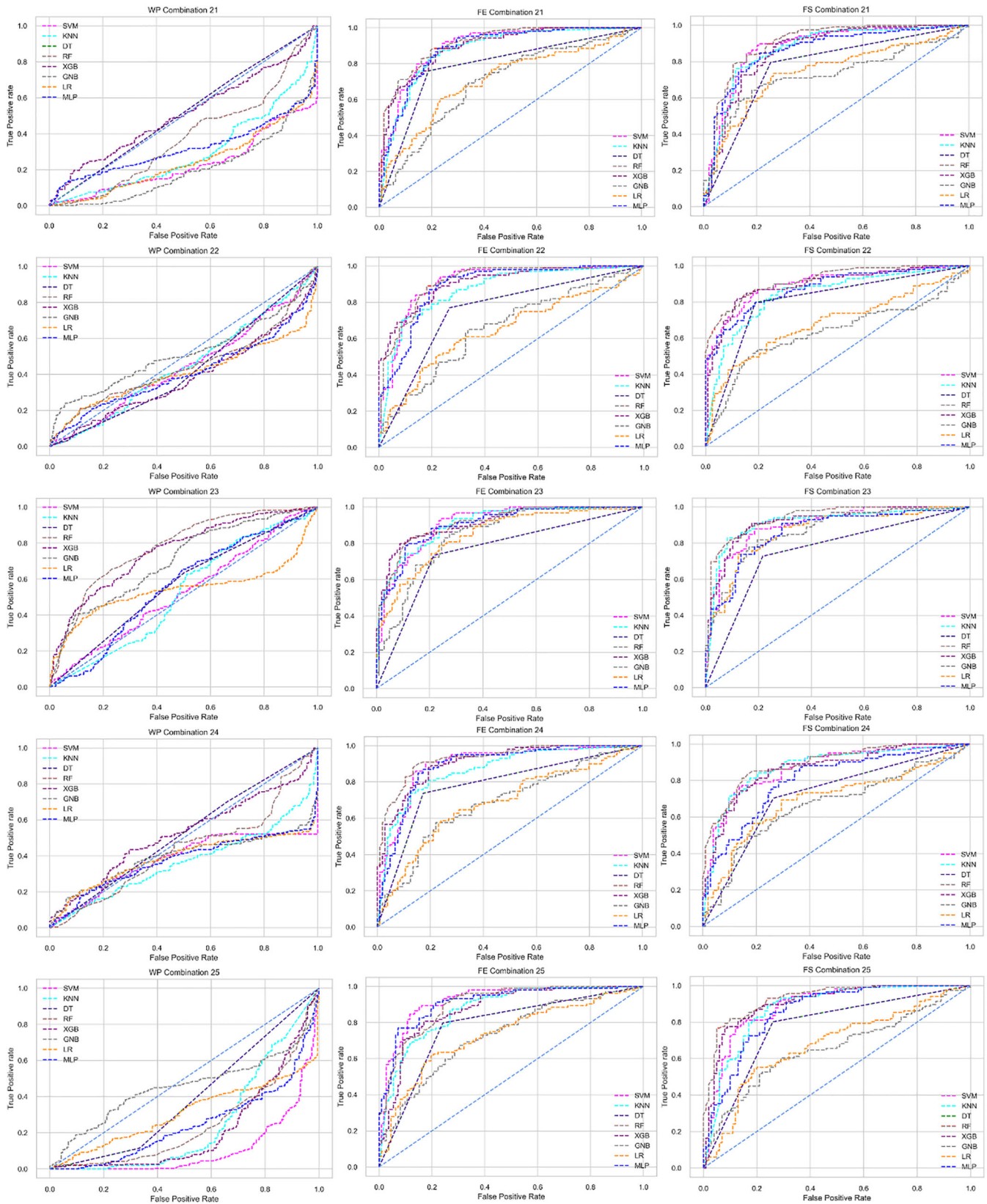

**Figure 19** ROC of the combination 2 (training) by 2 (testing).

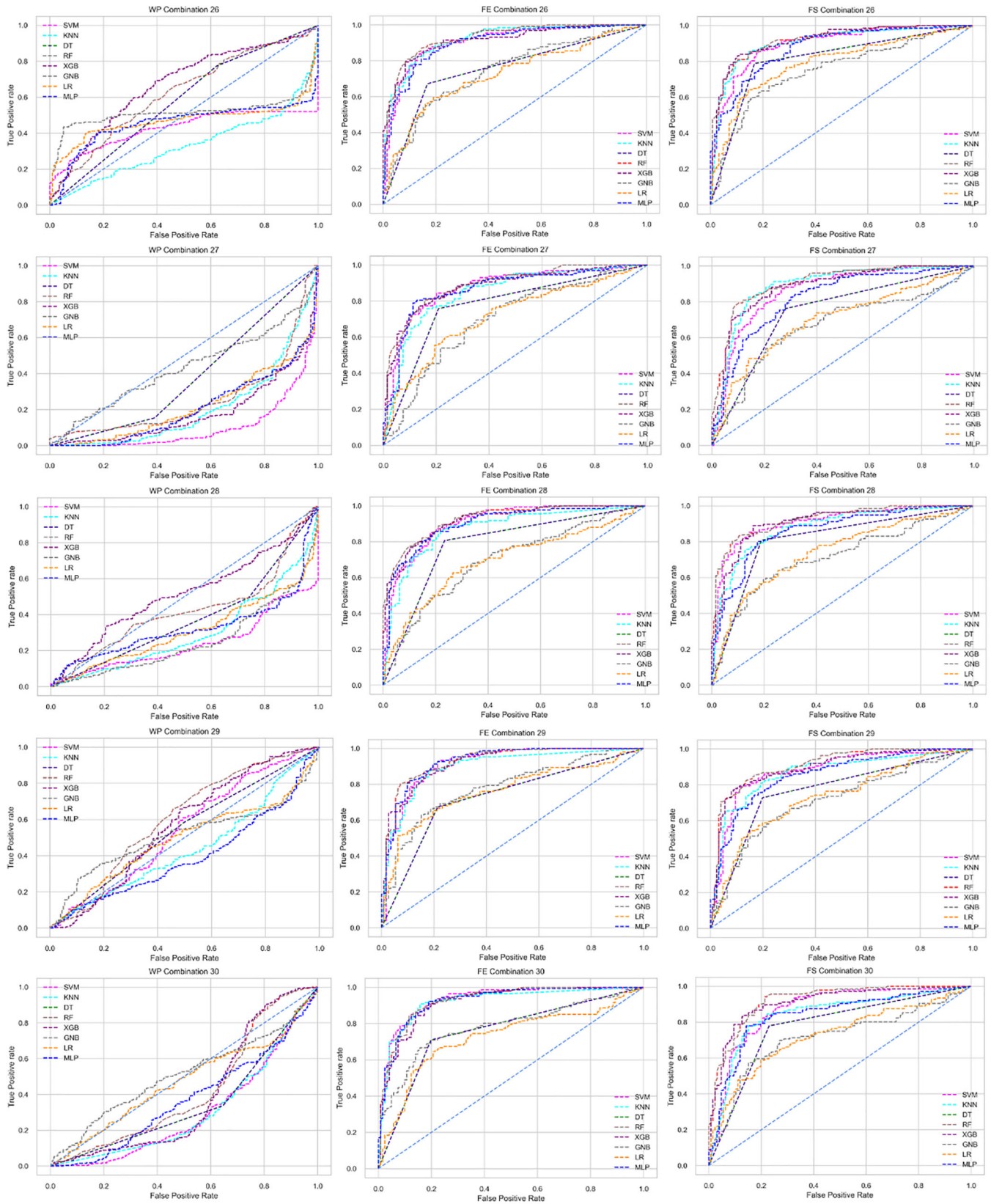

**Figure 20 ROC of the combination 3 (training) by 2 (testing).**

In Table 5, the combinations of 21 to 25 hold two datasets as training and two as testing phases. Here, also the classifiers show a low performance and unstable classification report. After applying the proposed preprocessing pipeline, the performance of the classifiers increases, indicating the mitigation of the inter-dataset performance discrepancy to predict heart disease. The performance of the classifiers is visualized in Fig. 19.

The last five combinations of Table 5 contain three datasets as training and two as testing phases. The performance of the classifiers is also low when we do not apply the proposed preprocessing pipeline; however, after applying the pipelines with PCA and RF, all of the classifier's performances increase, and the inter-dataset performance discrepancy mitigates. Also, all the ROC curves are in Fig. 20.

## Discussion

The main motivation of this study is to build a global ML model that can predict heart disease from any publicly available dataset. This model can also predict heart disease from multi-sensor data directly observed from the human body. After mitigating the inter-dataset performance discrepancy issue, the trained models are too flexible to other real-time data for prediction. To achieve this goal (inter-dataset performance discrepancy), we train different ML classifiers using the datasets and measure the performances of the classifiers. Besides this, we also train the classifiers using different combinations of the datasets, and in the testing phase, we also use different combinations of datasets. All the classifiers show unacceptable performances in the medical domain, where high accuracy is an urgent issue (*Sumathi & Poorna, 2016*). The problem behind the low performances of the classifiers is called the inter-dataset discrepancy problem. To mitigate this issue, we proposed a preprocessing pipeline where we apply both feature extraction and feature selection separately.

When we apply PCA as a feature extraction technique in the pipeline, the performance of the classifiers goes to a reasonable level. It sometimes rises to 95% accuracy, precision, and recall. Performances of the classifiers are stable, and the inter-dataset performance discrepancy problem solves most of the combinations. After applying RF in the preprocessing pipelines, the performances of the classifiers go high and stable. We get a good recall and specificity of the highly acceptable classifiers in this sensitive domain. Performances of the classifiers vary in different combinations. In some cases, it goes up; in others, it goes down. Among all the algorithms, RF is superior in predicting heart disease for individual datasets and inter-dataset setups. RF shows 96% accuracy with 96% precision, recall, and 99% AUC score. The results indicate that our proposed preprocessing pipeline solves the inter-datasets performance discrepancy, and the model is flexible to real-time observed data by sensors. We can use RF for preprocessing and classification purposes to solve this issue in heart disease prediction.

## CONCLUSION AND FUTURE WORK

This study introduced the inter-dataset performance discrepancy problem in five prevalent heart disease datasets and proposed a potential solution to mitigating this issue. The inter-dataset performance discrepancy problem arises due to the different statistical

characteristics of the datasets. Performances of the classifiers are low in the individual datasets and different training-testing combinations; however, high performance is one of the crucial facts in the medical domain of its sensitive characteristics. Inter-dataset performance discrepancy problem mitigation is one of the important issues in constructing a global dataset for any specific disease. So, we proposed a preprocessing pipeline that includes log transformation, outlier handling, data balancing, normalization, and dimensionality reduction against this issue. PCA as a feature extraction technique and RF as a feature selection technique is also integrated into the proposed pipeline. The results indicate our proposed pipeline mitigates the inter-dataset performance discrepancy problem in heart disease prediction. The performance of the classifiers goes to a high level, indicating the proof of our claim in this article. Moreover, RF outperforms feature selection and heart disease prediction in both the inter-dataset and single-dataset setups. The proposed method applies to secondary datasets with similar features. We should explore this domain more for datasets with dynamic updates and feature differences.

In the future, the proposed preprocessing pipeline could be used to minimize this type of issue for different chronic diseases, and it would help build a global model and global dataset for any specific disease. All datasets in this domain will be integrated to create a massive dataset and will be researched further. Since traditional ML methods cannot ensure data confidentiality, further studies could focus on privacy-preserving distributed computing.

### Funding
This research was supported by the Basic Science Research Program through the National Research Foundation of Korea (NRF) funded by the Ministry of Education (No. 2020R1I1A3069700) and by the Technology Development Program of MSS (No. S3033853). There was no additional external funding received for this study. The funders had no role in study design, data collection and analysis, decision to publish, or preparation of the manuscript.

### Grant Disclosures
The following grant information was disclosed by the authors:
Basic Science Research Program through the National Research Foundation of Korea (NRF) funded by the Ministry of Education: 2020R1I1A3069700.
Technology Development Program of MSS: S3033853.

### Competing Interests
The authors declare that they have no competing interests.

### Author Contributions
- Mahmudul Hasan conceived and designed the experiments, performed the computation work, prepared figures and/or tables, and approved the final draft.

- Md Abdus Sahid performed the experiments, performed the computation work, prepared figures and/or tables, and approved the final draft.
- Md Palash Uddin performed the experiments, authored or reviewed drafts of the article, and approved the final draft.
- Md Abu Marjan analyzed the data, authored or reviewed drafts of the article, and approved the final draft.
- Seifedine Kadry conceived and designed the experiments, prepared figures and/or tables, and approved the final draft.
- Jungeun Kim analyzed the data, authored or reviewed drafts of the article, and approved the final draft.

## Data Availability

The data and code are available in the Supplemental Files.

## Supplemental Information

Supplemental information for this article can be found online at http://dx.doi.org/10.7717/peerj-cs.1917#supplemental-information.

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
