# Peer review of "Performance discrepancy mitigation in heart disease prediction for multisensory inter-datasets"

_PeerJ Computer Science, doi:10.7717/peerj-cs.1917_

## Round 0.1 · original submission · Major Revisions

The article has some interesting content, but some flaws are present and need to be addressed. Please take care of them and prepare a new improved version of the manuscript.

**Language Note:** The review process has identified that the English language must be improved. PeerJ can provide language editing services - please contact us at copyediting@peerj.com for pricing (be sure to provide your manuscript number and title). Alternatively, you should make your own arrangements to improve the language quality and provide details in your response letter. – PeerJ Staff

Reviewer 1 ·

Basic reporting

The stucture of the paper looks good, it’s advised to be more succint.

However, the English of this paper needs major improvement.

In introducion,

Millions of people have attacked hearts every year -> Millions of people have had heaert attacks every year

Line 27, plaese add details to the sentence, "around 17.9 million people have died”, which year did that happen

Line 29, obsesity and the following aren’t “human behavior”

Line 32, please refhrase

Line 36, blackIn -> In

Line 781. Remove back

Experimental design

The dataset, and the steps of preprocessing, and the model structure are well documented. Data of the experiments are well documented and visualized.

Validity of the findings

The study combines five commonly used heart disease datasets—Cleveland, Hungary, Switzerland, Long Beach, and StatLog into one and does preprocessing and deduplication.

The novelty is extremely limited. Only existing ML methods are used, and no new model is proposed.

100% random forest is too high. It’d be more likely that the model overfits

Reviewer 2 ·

Basic reporting

The paper is generally well-structured with clearly defined sections, but there could be a more explicit statement of the research question or hypothesis.
Consider providing a brief summary of the main findings in the abstract to enhance its completeness.
The abstract provides a clear overview of the paper, outlining the significance of heart disease prediction and the research focus on inter-dataset discrepancy.
The introduction provides a comprehensive background on cardiovascular diseases and the role of machine learning in their prediction.

Experimental design

The paper describes the use of five datasets and their combination, demonstrating a thoughtful approach to addressing inter-dataset discrepancy.
Experimental design, including the use of various classifiers, preprocessing techniques, and performance assessment strategies, is well-detailed.
However, the paper lacks information on the selection criteria for the datasets and the rationale behind choosing specific preprocessin

Validity of the findings

The paper adequately discusses the potential impact of inter-dataset discrepancy and presents a pipeline to mitigate this issue.
The use of diverse classifiers, preprocessing techniques, and performance metrics strengthens the validity of the findings.
However, the paper could provide more details on the limitations of the study, potential biases in the datasets, and external factors that might affect the generalizability of the results.

Additional comments

The paper's writing is generally clear, but there are instances of repetition and verbose sentences. Consider refining the language for conciseness.
Ensure that all acronyms are defined upon first use for the benefit of readers unfamiliar with specific terms.
Figures and tables are referenced but not provided in the text. Including visual representations of experimental results could enhance the paper's clarity.
The paper lacks a clear statement of future work or recommendations for further research.
Overall, the paper addresses an important problem in heart disease prediction and presents a comprehensive experimental approach. Improvements in clarity, transparency in dataset selection, and addressing potential limitations would further enhance the quality of the paper.

---

## Round 0.2 · Minor Revisions

The authors correctly addressed most of the comments of the reviewers (who were invited to review this revision but did not do so), but some issues still remain. Among the binary classification outcomes, results should be measured through the Matthews correlation coefficient (MCC), negative predictive value (NPV), precision (PPV), Cohen's Kappa, and precision-recall curve AUC.

The discussion about the ROC curves is too long and misleading; ROC curves should not be highlighted ( https://doi.org/10.1111/j.1466-8238.2007.00358.x and https://doi.org/10.1007/s00357-019-09345-1 ).

Moreover, plots regarding the dimensionality reduction should be included.

---

## Round 0.3 · Minor Revisions

The authors refused to address my request by not including results measure through Matthews correlation coefficient (MCC) and negative predictive value (NPV). This article cannot be accepted for publication by me without them..

I am giving you one final chance to provide the required measures.

---

## Round 0.4 · accepted · Accept

The authors addressed my requests ans therefore I can recommend this article for acceptance.